# Frictional instabilities as an alternative to friction coefficient in fine touch perception

**Maryanne Derkaloustian[1], Pushpita Bhattacharyya[2], Truc T Ngo[3], Josh GA Cashaback[3], Jared Medina[2,4], Charles B Dhong[1,3]***

[1]Department of Materials Science and Engineering, University of Delaware, Newark, United States; [2]Department of Psychological and Brain Sciences, University of Delaware, Newark, United States; [3]Department of Biomedical Engineering, University of Delaware, Newark, United States; [4]Department of Psychology, Emory University, Atlanta, United States

**\*For correspondence:**
cdhong@udel.edu

**Competing interest:** The authors declare that no competing interests exist.

## eLife Assessment

This **useful** study integrates experimental methods from materials science with psychophysical methods to investigate how frictional stabilities influence tactile surface discrimination. The authors argue that force fluctuations arising from transitions between frictional sliding conditions facilitate the discrimination of surfaces with similar friction coefficients. However, the reliance on friction data from an artificial finger, combined with correlational analyses that fall short of establishing a mechanistic link to perception, renders the findings **incomplete**.

**Abstract** Fine touch perception is often correlated to material properties and friction coefficients, but the inherent variability of human motion has led to low correlations and contradictory findings. Instead, we hypothesized that humans use frictional instabilities to discriminate between objects. Here, we constructed a set of coated surfaces with minimal physical differences, but due to differences in surface chemistry, generated different types of instabilities depending on how quickly a finger is slid and pressed during sliding. In one experiment, we used a mechanical mock finger to quantify and classify differences in instability formation from different coated surfaces. In a second experiment, participants perform a discrimination task using the same coated surfaces. Using the data from these two experiments, we found that human discrimination response times were faster with surfaces where the mock finger produced more stiction spikes and discrimination accuracy was higher where the mock finger produced more steady sliding. Conversely, traditional metrics like surface roughness or average friction coefficient did not relate to tactile discriminability. In fact, the typical method of averaging friction coefficients led to a spurious correlation which erroneously suggests that distinct objects should feel identical and identical objects should feel distinct—similar to findings by others. Friction instabilities may offer a more predictive and tractable framework of fine touch perception than friction coefficients, which would accelerate the design of tactile interfaces.

## Introduction

People can readily distinguish and describe a variety of textures by touch, but there is no single material parameter or property that consistently predicts human fine touch. Fine touch arises from the friction generated between a finger and an object, so the most common method to classify objects in

fine touch is by assigning a constant friction coefficient to different materials (*Smith and Scott, 1996*; *Khamis et al., 2021*). Although the friction coefficient is often convenient, it is not a material property (*Ben-David and Fineberg, 2011*) and most findings have yielded either contradictory results or low statistical correlations to human performance, like discriminability or identification (*Khamis et al., 2021*; *Skedung et al., 2020*; *Zhou et al., 2020*; *Olczak et al., 2018*). The cause of these discrepant findings is, in part, that sliding friction in soft systems has significant oscillations in forces arising from microscale stick-slip, adhesion, and elasticity (*Rand and Crosby, 2006*). Indeed, the inherent human variability in finger velocity and pressure means that the variations in friction encountered on a single object can be larger than the variations in friction between two distinct objects (*Nolin et al., 2021*; *Nolin et al., 2022*; *Peng et al., 2021*).

Instead of a material property or parameter, we propose that humans can tell surfaces apart due to the likelihood of encountering frictional instabilities like stiction spikes and slow friction waves. These frictional instabilities arise due to the competition between the elasticity of the finger and adhesion to a substrate (*Rand and Crosby, 2006*). In an experimental parameter space of finger pressure and velocity, the boundary where one instability type forms versus another is dependent upon surface composition (*Nolin et al., 2022*; *Putelat et al., 2017*). We surmise that humans actively explore surfaces to find these transitions between different types of frictional instabilities to tell surfaces apart. Classifying surfaces based on transitions between instabilities would be advantageous because it is less variable than raw friction forces typically seen in macroscopic experiments (*Putelat et al., 2017*).

Most friction at solid interfaces is characterized by some degree of stick-slip behavior, where the fluctuations in force can largely depend upon sliding conditions as well as the material properties of each object (*Rice and Ruina, 1983*; *Shoaib et al., 2018*). In elastic, rigid systems, a critical spring constant dictated by sliding conditions and slip history can determine whether steady sliding or different forms of slip instabilities occur (*Rice and Ruina, 1983*; *Ruina, 1983*). These rate and state laws gain even more complexity when one object is soft and deformable and also makes adhesive contact with a hard surface. Prior to any slip events, the effects of contact aging are amplified with significant intermolecular forces at the interface (*Petrova et al., 2020*), and the drastic evolution of contact area while sliding largely controls the resulting friction force (*Sahli et al., 2018*). Additionally, local tensile and compressive stresses can often create periodic oscillations in this force, even in the absence of mesoscale interface detachment (*Viswanathan et al., 2016a*). These phenomena are sufficiently explained for controlled systems with a constant normal load and known initial conditions, but the dynamics arising from the unconstrained nature of human touch and their mechanistic origins can be more difficult to predict.

Due to these systematic variations in friction, deriving relationships between a constant or average friction coefficient and tactile perception has led to low correlations or findings that do not generalize between studies. Typically, psychophysical studies using this approach rely on combining friction coefficients with other material properties and tactile dimensions defined by qualitative descriptors to extract correlations to participants' ability to distinguish between surfaces (*Skedung et al., 2020*; *Dacleu Ndengue et al., 2017*). Instead, more direct correlations have often emerged from studies investigating dynamic friction during touch (*Willemet et al., 2021*; *Schwarz, 2016*). *Gueorguiev et al., 2016* identify a partial slip region, where a finger's peripheral contact area begins to drop until a threshold of smoother sliding, as a strong tactile cue in discriminating between glass and PMMA (acrylic) slabs. However, the connections between their significant differences in physiochemical properties and their stick-slip behavior at the mesoscale remain unclear as these partial slips are dependent on finger velocity and pressure, and human fingers vary in both during free exploration.

Here, we investigate dynamic friction as a predictor across controlled surfaces to establish structure-dynamics-property relationships connecting molecular scale phenomena to mesoscale mechanics. Within the ranges of applied finger pressures and sliding velocities observed in humans, the same material can exhibit multiple variations of stick-slip phenomena (*Nolin et al., 2021*; *Nolin et al., 2022*; *Swain et al., 2024*), which challenges correlations derived from singular measurements of tangential force.

# Materials and methods

## Mock finger as a characterization tool

We use a mechanical setup with a PDMS (poly(dimethylsiloxane)) mock finger to derive tactile predictors as opposed to direct biomechanical measurements on human participants. While there is a tradeoff in selecting a synthetic finger over a real human finger to modeling human touch, human fingers themselves are also highly variable (*Infante et al., 2025*) both in their physical shape and their use during human motion. Our goal is to design a consistent method of characterization of samples that can be easily accessed by other researchers and does not rely on a standard established around single human participants. We believe that sufficient replication of surface, bulk properties, and contact geometry results in characterization that isolates consistent features of surfaces that are not derived from human-to-human variability. We have used this approach to successfully correlate human results with mock finger characterization previously (*Nolin et al., 2021*; *Nolin et al., 2022*; *Nolin et al., 2024*).

The major component of a human finger, by volume, is soft tissue (~56%) (*Murai et al., 1997*), resulting in an effective modulus close to 100 kPa (*Abdouni et al., 2017*; *Cornuault et al., 2015*). In order to achieve this same softness, we crosslink PDMS in a 1×1×5 cm mold at a 30:1 elastomer:crosslinker ratio. In addition, two more features in the human finger impart significant mechanical differences. Human fingers have a bone at the fingertip, the distal phalanx (*Abdouni et al., 2017*; *Cornuault et al., 2015*; *Qian et al., 2014*), which we mimic with an acrylic 'bone' within our PDMS network. The stratum corneum, the stiffer, glassier outer layer of skin (*Yuan and Verma, 2006*), is replicated with the surface of the mock finger glassified, or further crosslinked, after 8 hr of UV-Ozone treatment (*Fu et al., 2010*). This treatment also modifies the surface properties of the native PDMS to align with those of a human finger more closely: it minimizes the viscoelastic tack at the surface, resulting in a comparable non-sticky surface. Stabilizing after one day after treatment, the mock finger surface obtains a moderate hydrophilicity (~60°), as is typically observed for a real finger (*Dhong et al., 2018*).

The initial contact area formed before a friction trace is collected is a rectangle of 1×1 cm. While this shape is not entirely representative of a human finger with curves and ridges, human fingers flatten out enough to reduce the effects of curvature with even very light pressures (*Dhong et al., 2018*; *Israelachvili, 2011*; *Gao et al., 2004*). This implies that for most realistic finger pressures, the contact area is largely load-independent, which is more accurately replicated with a rectangular mock finger.

Lastly, we consider the role of fingerprint ridges. A key finding of our previous work is that while fingerprints enhanced frictional dynamics at certain conditions, key features were still maintained with a flat finger (*Dhong et al., 2018*). Furthermore, for some loading conditions, the more amplified signals could also result in more similar friction traces for different surfaces. We have observed good agreement between these friction traces and human experiments (*Nolin et al., 2021*; *Nolin et al., 2022*; *Swain et al., 2024*; *Carpenter et al., 2018*).

## Surface preparation

Silane coatings were prepared via chemical vapor deposition (CVD) onto 10-cm-diameter silicon wafers (University Wafer). Clean wafers were first treated with oxygen plasma (Glow Research Glow Plasma System) for 1 min. They were immediately moved to a desiccator containing ~50 µL of silane (Gelest, 95–100% purity) on a glass slide. Alkylsilanes and aminosilanes each required a separate desiccator. After pulling vacuum, each desiccator was held under static vacuum for at least 4 hr. Evidence of plasma treatment as well as deposition was confirmed with atomic force microscopy (AFM), x-ray photoelectric spectroscopy (XPS), and water contact angle hysteresis experiments.

## Atomic force microscopy

1×1 cm wafer pieces were imaged with a Bruker Multimode AFM to compare surface profiles of all silanes as well as the effects of plasma treatment. 10×10 µm scans were generated under tapping mode at a frequency of 1 Hz with tips of a~120 kHz resonant frequency, and analyzed with Gwyddion microscopy data analysis and visualization software. All height profiles were processed with a third-order background subtraction, a median height correction along the fast scanning (*x*) axis, and

horizontal stroke correction prior to analysis. Aggregates were not masked as they only negligibly changed power spectral density profiles.

## X-ray photoelectric spectroscopy

To identify the elements present on our surfaces, 1×1 cm samples were scanned with a Thermo Scientific K-Alpha XPS system. 400 μm points were scanned on each sample, for individual elements C, N, O, and Cl, as well as an XPS survey scan.

## Water contact angle hysteresis

Advancing and receding angles of water droplets on our surfaces were measured using a goniometer (DSA14 Drop Shape Analysis System, Kruss). To measure advancing angles, ~3 μL of DI water was dispensed onto each surface, and an image of the droplet-surface interface was captured. ~1 μL of water was then slowly removed from the droplet such that it would recede, when another image was captured. Each droplet was fit to an automatic circle fit in ImageJ to obtain advancing and receding angles. This was repeated five times for each surface, in order to obtain average angles and average hysteresis by subtracting the two.

## Mock finger preparation

Friction forces across all six surfaces were measured using a custom apparatus with a polydimethylsiloxane (PDMS, Dow Sylgard 184) mock finger that mimics a human finger's mechanical properties and contact mechanics while exploring a surface relatively closely (*Nolin et al., 2021*; *Nolin et al., 2022*). PDMS and crosslinker were combined in a 30:1 ratio to achieve a stiffness of 100 kPa comparable to a real finger, then degassed in a vacuum desiccator for 30 min. We are aware that the manufacturer-recommended crosslinking ratio for Sylgard 184 is 10:1 due to potential uncrosslinked liquid residues (*Glover et al., 2020*), but further crosslinking concentrated at the surface prevents this. The prepared PDMS was then poured into a 1×1×5 cm mold also containing an acrylic 3D-printed 'bone' to attach applied masses on top of the 'fingertip' area contacting a surface during friction testing. After crosslinking in the mold at 60°C for 1 hr, the finger was treated with UV-Ozone for 8 hr out of the mold to minimize viscoelastic tack.

## Mechanical testing

A custom device using our PDMS mock finger was used to collect macroscopic friction force traces replicating human exploration (*Nolin et al., 2021*; *Nolin et al., 2022*). After placing a sample surface on a stage, the finger was lowered at a slight angle such that an initial 1×1 cm rectangle of 'fingertip' contact area could be established. We considered a broad range of applied masses ($M$=0, 25, 75, and 100 g) added onto the deadweight of the finger (6 g) observed during a tactile discrimination task. The other side of the sensor was connected to a motorized stage (V-508 PIMag Precision Linear Stage, Physik Instrumente) to control both displacement (4 mm across all conditions) and sliding velocity ($v$=5, 10, 25, and 45 mm s$^{-1}$). Forces were measured at all 16 combinations of mass and velocity via a 250 g Futek force sensor ($k$=13.9 kN m$^{-1}$) threaded to the bone, and recorded at an average sampling rate of 550 Hz with a Keithley 7510 DMM digitized multimeter. Force traces were collected in sets of four slides, discarding the first due to contact aging. Because some mass-velocity combinations were near the boundaries of instability phase transitions, not all force traces at these given conditions exhibited similar profiles. Thus, three sets were collected on fresh spots for each condition to observe enough occurrences of multiple instabilities, at a total of nine traces per combination for each surface.

## Instability classification

Without any previously defined classifications of the instabilities in our systems, we restricted our force traces to 3 main types discussed in the main text: steady sliding, slow frictional waves, or stiction spikes. However, some stiction spikes were not immediately categorizable by eye, as performed independently by two coauthors, but consistency was maintained depending on whether these spikes were followed by steady sliding or slow frictional waves.

## Pair selection for human testing

With nine friction force traces for each mass-velocity pair and surface, large frequencies of instabilities could be counted and used to generate instability phase maps for each material. Frequencies of

**Table 1.** Surface characterization of silanized silicon wafers.

| Sample | Surface structure | Average roughness ($R_a$, pm) | Contact angle (advancing - receding ± $\sigma_{Hysteresis}$, °) | Average friction coefficient ($\bar{\mu}$) | Friction coefficient interquartile range ($\mu_{Q3} - \mu_{Q1}$) | Hurst exponent ($H$) |
|---|---|---|---|---|---|---|
| C4 | | 861 | (99.06–103.74)±2.5 | 2.21 | 1.24 | 0.56 |
| C5 | | 443 | (100.18–109.68)±2.6 | 2.45 | 2.09 | 0.22 |
| C6 | | 858 | (102.42–109.10)±5.1 | 2.49 | 1.74 | 0.29 |
| C8 | | 431 | (108.38–114.32)±4.0 | 3.04 | 2.08 | 0.06 |
| C4-APTMS | | 358 | (54.74–59.44)±3.9 | 2.06 | 1.20 | 0.04 |
| APTMS | | 280 | (49.80–58.28)±5.7 | 2.03 | 1.54 | 0.15 |

occurrence of steady sliding, stiction spikes, and slow frictional waves on each surface were determined manually, as well as total incidence of instabilities. These frequencies corresponded to color intensities for each instability on the phase maps, revealing similar zones but different boundaries for every surface.

The differences between these frequencies were calculated for several pairs, both previously tested and new pairs. Six pairs with a wide range of differences in total instabilities were selected, three of which had been tested prior to this work. These pairs had different ranges of individual instability differences as well, as shown in Table 2.

### Human testing

To verify that the differences in instabilities aid in tactile discrimination, we performed three-alternative forced choice tests (3-AFC) (N=10 participants, n=600 trials) on 10 sighted adult participants (IRB approval in Acknowledgments). In each trial, subjects were presented with three surfaces lined up horizontally to touch, two having the same silane coating and an odd sample out. Pairs as well as unique sample placement were randomized across all subjects and trials. No blindfolds or visual barriers were used since the appearances of all coatings were the same. Participants could touch for as long as they wanted, but were asked to only use their dominant index fingers along a single axis to better mimic the conditions for instability formation during mechanical testing with the mock finger. Once a participant made a final decision on which sample was unlike the other two, we input their answer in a premade Qualtrics form which recorded the time to the last click as the response time.

## Results and discussion

### Generating phase maps of frictional instabilities

To test the hypothesis that humans use frictional instabilities to form tactile judgements, we constructed 'minimal' tactile surfaces made from the vapor deposition of silanes onto silicon wafers for human testing (*Table 1*).

We describe these samples as 'minimal' because the physical variations in roughness are below the human limits of detection (*Skedung et al., 2013*; *AliAbbasi et al., 2023*) and the bulk and thermal properties of all samples are identical at the human scale because the silanes form a thin layer of order nanometers thickness (*Wasserman et al., 1989*; *Zhu et al., 2012*). Thus, the only present differences in these samples are from those in surface chemistry (*Table 1*, see Appendix for details). We propose

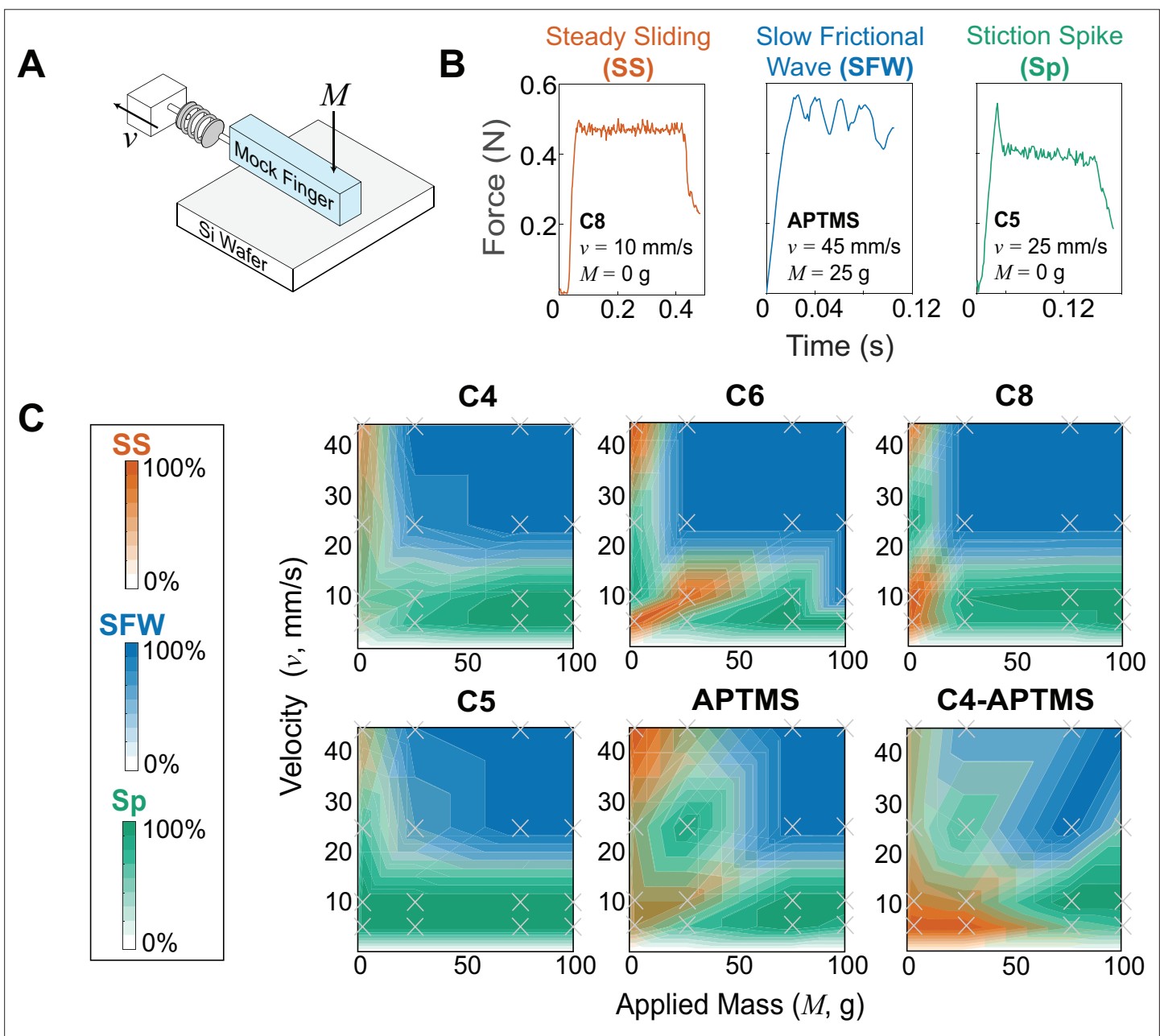

**Figure 1.** Mechanical testing and determination of frictional instability landscape in samples. (**A**) Schematic of friction testing apparatus. An elastomeric mock finger is connected to a force sensor and motorized stage. The mock finger was driven at four different driving velocities (v=5–45 mm/s) at four different applied masses (M=0–100 g, in addition to the deadweight of the mock finger), mimicking human exploration. (**B**) Representative force curves of three types of frictional instabilities. *Steady sliding* (**SS**, orange) shows no distinctive feature besides high frequency force oscillations due to stick-slip, which is above the electronic sensor noise. *Slow frictional waves* (**SFW**, blue) are slower, large amplitude oscillations corresponding to global or coherent stick-slip of the entire mock finger. *Stiction spike* (**Sp**, green) is where a single, large-magnitude stick-slip event is observed at the onset of steady sliding. (**C**) Phase maps of frictional instabilities across all surfaces. Colors and intensities correspond to the type and frequency of each instability, respectively, at a given condition. Color map created by linear interpolation between experimental conditions, with location of conditions tested indicated by gray (×) marks.

that the role of these differences in surface chemistry is to shift the boundaries between frictional instabilities to different finger pressures and velocities. As there is no existing method to accurately predict phase maps of instabilities of elastic bodies from molecular structure, we experimentally determined these frictional instability maps through a mechanical apparatus with a mock finger (*Figure 1*).

Across the six different surface coatings and 16 different combinations of finger sliding velocity and applied mass, we observed that the friction traces could be categorized into three broad phases, which conceptually originate from the competition between the adhesion and elasticity of the finger. The first phase, *steady sliding* (**SS,** orange in *Figure 1B*), shows a nearly constant friction force, except for high-frequency mechanical oscillations (*Israelachvili, 2011*; *Das et al., 2015*). Mechanistically, this represents sufficient dissipation of kinetic energy relative to driving velocity, thus there is a lack of macroscale or concerted stick events that lead to large oscillations in force (*Gueorguiev et al., 2016*; *Israelachvili, 2011*). Instead, microscale stick-slip events result in a high-frequency, but small, oscillation around a comparatively steady mean. We observe that this occurs primarily at low applied masses, but different velocity zones depending on the surface. The initial slope in the steady sliding regime, and in all other phases, is equal to $k_{finger}$. The second phase, *slow frictional waves* (**SFW,** blue in *Figure 1B*), manifest as large oscillations in friction force with frequencies lower than any representative timescale for microscopic stick-slip or motor velocity. These oscillations are much larger than electronic noise of ~0.04 N. Slow frictional waves represent a scenario where kinetic energy by friction and adhesion is not dissipated sufficiently quick. Instead, stresses within the mock finger can localize to form instabilities like buckling (*Briggs and Briscoe, 1978*), pulses (*Baumberger et al., 2002*), or wrinkling (*Rand and Crosby, 2009*) to relieve the buildup of elastic stresses. We observe that slow frictional waves occurred more frequently at higher masses and velocities compared to steady sliding. This is expected because higher normal loads increase the tangential forces required for motion, and the higher velocities give shorter times to resolve stress irregularities (*Viswanathan et al., 2016b*). We also observed a third phase, a singular *stiction spike* (**Sp,** green in *Figure 1B*), that precedes an otherwise smooth, or if also exhibiting a slow frictional wave, comparatively small oscillating trace. This stiction spike represents where the initial barrier to motion, adhesion, represents a significant threshold above the mean sliding friction (*Persson et al., 2004*), likely because most of these microscale contacts are broken within a short interval. Stiction spikes were routinely observed even after we discarded the first pull out of a sequential series of four pulls to remove the influence of contact aging originating from setup preparation. Stiction spikes are not necessarily mutually exclusive to steady sliding and slow frictional waves and can coexist with the others (see *Appendix 1— figure 4* of Appendix). However, based on our heuristic classification scheme, a 10% higher spike than the mean during steady sliding was considered a stiction spike trace. When the spike was followed by slow frictional waves, the traces with the first peak being 40% higher in amplitude were also classified as stiction spikes.

In many cases, we observed that several trials in the same conditions consistently led to mixed phases, despite initially assuming that these originated from experimental error. Upon construction of the phase maps, we saw that mixed phases occur predominantly at boundaries. Conversely, in the interior, we see higher consistency (darker shade). Generally, we see that steady sliding is less frequent in short-chained, unaligned surfaces like **C4** and **C5** (see Hurst exponents in *Table 1*). Although no analytical modeling of soft, mesoscale friction analysis has provided accurate phase maps of these instabilities, theoretical work by *Putelat et al., 2017* and experimental work by others *Viswanathan et al., 2016a*; *Maksuta et al., 2022* have shown that these instability phase maps can have abrupt transitions.

As *Figure 1* was constructed from friction measurements, we can also calculate an average friction coefficient, $\bar{\mu}$, by averaging the friction coefficient obtained at each of the 16 combinations of masses and velocities (*Table 1*). This calculation is a standard approach in tactile studies for summarizing friction measurements, or in some cases, surfaces are never characterized at multiple masses and velocities. However, summarizing friction data in this manner has been considered conceptually questionable by others from a mechanics perspective (*Ben-David and Fineberg, 2011*). *Figure 1* shows that the type of instabilities and friction forces encountered on a single surface can vary widely depending on the conditions. As a result, large variations in the friction coefficient are expected, depending on the mass and velocity — even though measurements originate from the same surface. This variability in friction coefficient can be seen with the large interquartile range of friction coefficients, which shows that the

**Table 2.** Pairs of samples for human testing by their frequency of instability formation.

| Label | Sample comparison | |ΔSS| | | |ΔSFW| | | |ΔSp| | |
|---|---|---|---|---|---|---|---|
| | | Count | % | Count | % | Count | % |
| P1 | C4 *vs* C4-APTMS | 20 | 13.9% | 16 | 11.1% | 4 | 2.78% |
| P2 | C6 *vs* C4-APTMS | 9 | 6.25% | 27 | 18.8% | 14 | 9.72% |
| P3 | C4 *vs* C5 | 11 | 7.64% | 21 | 14.6% | 14 | 9.72% |
| P4 | C8 *vs* C4-APTMS | 14 | 9.72% | 34 | 23.6% | 2 | 1.39% |
| P5 | C4 *vs* APTMS | 24 | 16.7% | 16 | 11.1% | 8 | 5.56% |
| P6 | C5 *vs* C4-APTMS | 31 | 21.5% | 23 | 16.0% | 18 | 12.5% |

variation in friction coefficient across a single surface is similar, or even larger, than the differences in average friction coefficient across two different surfaces. The observation that friction coefficients vary so widely on a single surface calls into question the approach of analyzing how humans may perceive two different objects based on their average friction coefficients.

## Human participants testing

The results from *Figure 1* demonstrate that we can design human experiments with different pairs of surfaces with differential frequencies in the occurrence of instabilities. To determine if humans can detect these three different instabilities, we selected six pairs of surfaces to create a broad range of potential instabilities present across all three types. These are summarized in *Table 2*, where the first column for each instability is the difference in that instability formed between each pair, and the second is the percent difference. The percentages are determined by, in an example, **C4** has steady sliding in 23 out of 144 mechanical pulls, and **C4-APTMS** has steady sliding occurring in 43 pulls. Thus, when comparing **C4** versus **C4-APTMS**, they have a difference in steady sliding of 20 out of a maximum 144 pulls, for a |ΔSS| of 13.9%. The absolute value is taken to compare total differences present, as the psychophysical task does not distinguish between sample order.

10 participants were asked to perform a three alternative-forced choice task (3-AFC, n=600 total trials), where they are given three samples at a time, two of which have the same surface coating, and one of which has a different coating — the identity and placement of the 'odd-man out' sample was randomized. Participants were asked to select which of the one coatings is unlike the other, which is advantageous because participants were not prompted to select based on potentially subjective percepts, such as which one feels 'smoother' or 'rougher' (*Kingdom and Prins, 2016*). To prevent any confounding effects from surface fouling by finger residue (*Swain et al., 2024*; *Xiao et al., 1995*), all samples were used only a single time. Blindfolds were unnecessary as all samples are visually identical. Additional testing details are provided in Materials and methods.

Participants were successful in distinguishing between all pairs (*Figure 2A*), with all average accuracies above chance (33%, p<0.005 by one-sample *t*-test). There were statistically significant differences in performance across some pairs (p<0.05, Wilcoxon rank sums) as indicated in *Figure 2A*. Pairs that had similar accuracies are still insightful: as a conceptual example, participants had nearly the same accuracy with *P1* and *P6*, but from *Table 2*, *P1* had small differences (2.78%) in stiction spike production, whereas *P6* had large differences (12.50%).

We used the differential frequencies of each instability in *Table 2* to perform a statistical fit of participant results by using a generalized linear mixed model (GLMM, details in Appendix). Considering all instabilities individually, we found that only steady sliding was a positive, statistically significant predictor (r=0.62, p<0.05, shown in *Figure 2B*). A positive correlation means that the more differences in steady sliding formation between a pair of material lead to higher accuracy in discrimination. Multi-term models including multiple instabilities were not statistically significant. Furthermore, a model accounting for slow frictional waves alone specifically shows a significant, negative effect on performance (p<0.01, *Appendix 1—figure 5* of Appendix), suggesting that in these samples and task, the type of instability was not as important. Thus, forming either type of instability (slow frictional waves or stiction spikes) on one surface compared to another surface which consistently generates steady sliding will lead to two surfaces feeling very distinctive.

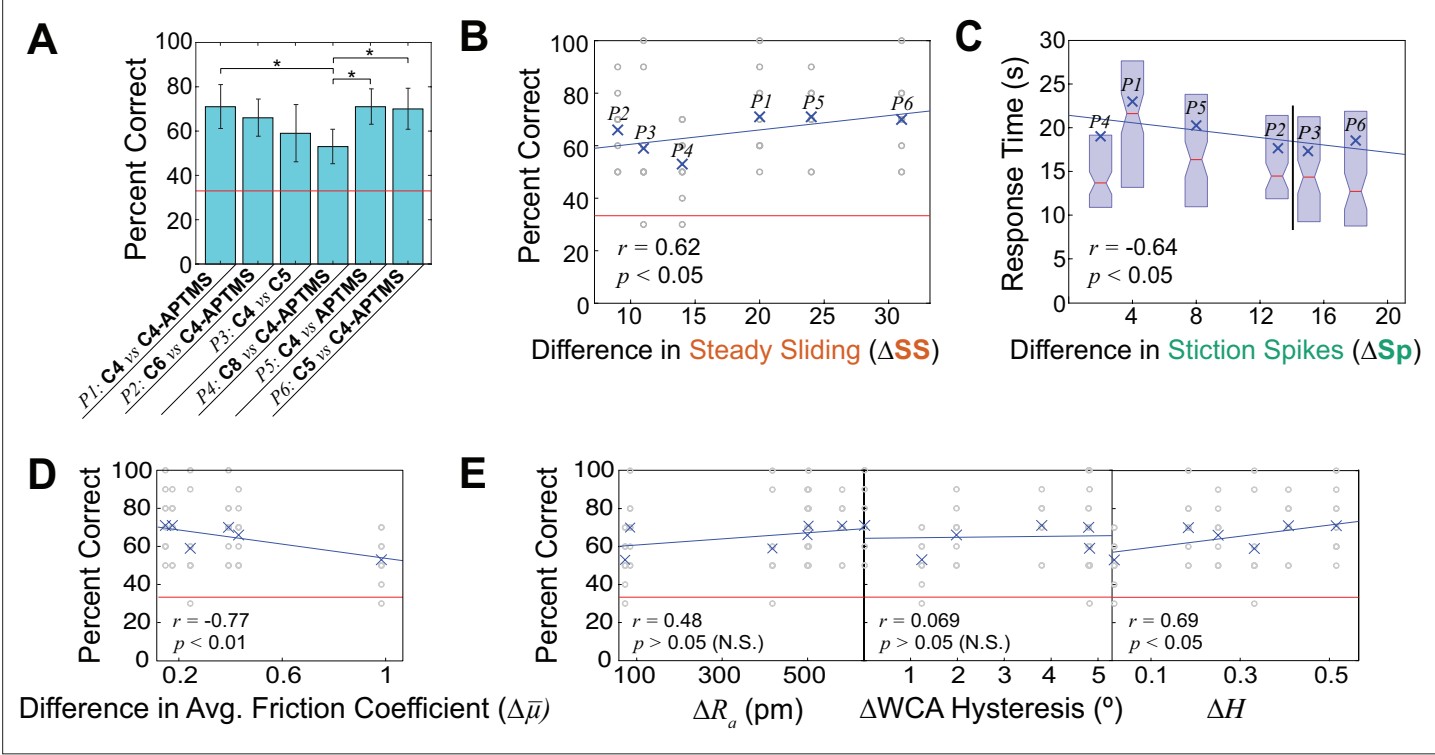

**Figure 2.** Human subjects testing for discriminability between surfaces. (**A**) Average accuracies of all pairs in a 3-AFC task. All pairs were statistically significantly distinguishable above chance (red line, 33%). Error bars are represented as 95% confidence intervals around the means. Ends of brackets with (*) above denote statistically significantly different accuracy rates. (**B**) Generalized linear mixed model (GLMM) of human discrimination accuracy during fine touch vs. the differences in steady sliding events determined by the mock finger. Accuracies of each trial across all subjects represented by gray O marks, and global averages represented by blue × marks. (**C**) GLMM of response times vs. the differences in stiction spike events. *P2* and *P3* have the same x-axis value and are shifted for clarity. Mean times are represented by blue (×) marks, while medians are represented by red lines at notches of box plots. (**D**) GLMM of accuracy vs. difference in average friction coefficient ($\Delta\bar{\mu}$), showing a negative correlation. (**E**) GLMMs of accuracy vs. other commonly used material properties or parameters: ΔAverage roughness $R_a$, ΔHurst exponent $H$, and ΔWater contact angle hysteresis (°) (N=10 participants, n=600 total trials).

Average response times for each pair were also fit to a GLMM to determine if the type of instability affected decision-making speeds. Participants were quicker to discriminate between surfaces if greater differences in stiction spikes were produced on the mock finger (p<0.05, shown in *Figure 2C*), while there was no significant relationship between steady sliding or slow frictional waves on response times. This may be because stiction spikes are the most prevalent instability in all surfaces except **C6**, so occurrences of steady sliding or slow frictional waves against a background of stiction spikes may be important in forming tactile judgements. Supporting the idea that deviances from stiction spike occurrence were important, we found that they were not a predictor of subject accuracy, only response time.

To compare the value of looking at frictional instabilities, we also performed GLMM fits on common approaches in the field, like a friction coefficient or material property typically used in tactile discrimination, shown in *Figure 2D–E*. Interestingly, in *Figure 2D*, we observed a spurious, negative correlation between friction coefficient (typically and often problematically simplified as $F = \bar{\mu}N$ across all tested conditions) and accuracy (r=–0.64, p<0.01); that is, the more different the surfaces are by friction coefficient, the less people can tell them apart. This spurious correlation would be the opposite of intuition and further calls into question the common practice of using friction coefficients in touch-related studies. Interestingly, this spurious correlation was also found by *Gueorguiev et al., 2016* The alternative, two-term model which includes adhesive contact area for friction coefficient (*Israelachvili, 2011*) was even less predictive (see *Appendix 1—figure 6A* of Appendix).

We investigate different material properties in *Figure 2E*. Differences in average roughness $R_a$ (or other parameters, like root mean square roughness $R_{rms}$ *Appendix 1—figure 6A* of Appendix) did

not show a statistically significant correlation to accuracy. Though roughness is a popular parameter, correlating any roughness parameter to human performance here could be moot: the limit of detecting roughness differences has previously been defined as 13 nm on structured surfaces (*Skedung et al., 2013*) and much higher for randomly rough surfaces (*Sahli et al., 2020*), all of which are magnitudes larger than the roughness differences between our surfaces. The differences in contact angle hysteresis – as an approximation of the adhesion contributions (*Chen et al., 1991*) – do not present any statistically significant effects on performance.

Interestingly, a large and statistically significant correlation is observed with the Hurst exponent *H* (p<0.05), indicating differences in monolayer ordering are also likely to create distinct tactile feedback. This was expected since our silane-derived coatings cause frictional differences due to monolayer ordering (*Lio et al., 1997*; *Xiao et al., 1996*). However, the Hurst exponent was positively correlated with response time (see *Appendix 1—figure 6B* of Appendix), indicating that bigger differences in Hurst led to slower decision times, which is counterintuitive. Thus, while the Hurst

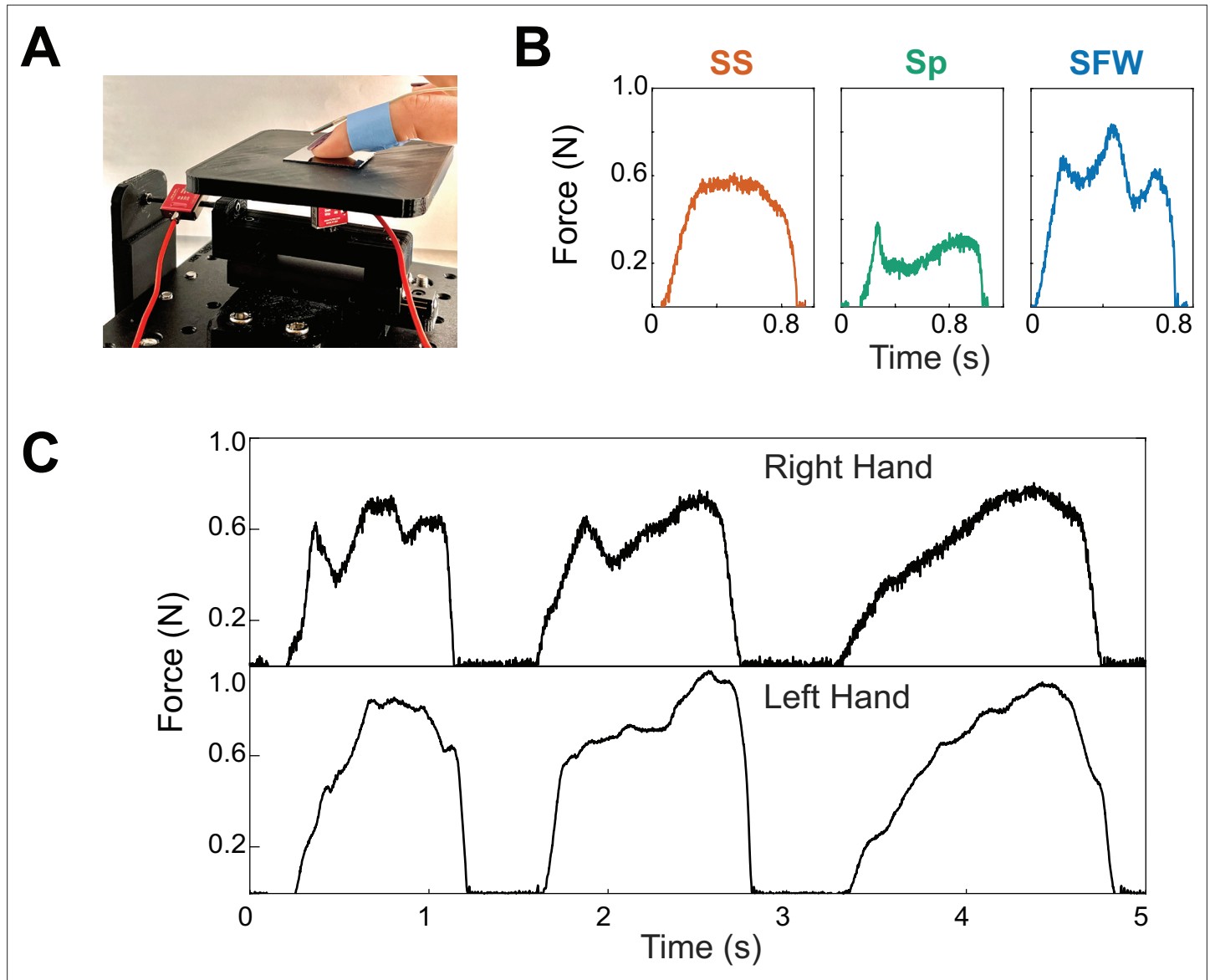

**Figure 3.** Recording friction of a human finger exploring a surface. (**A**) Setup to measure tangential and normal forces during fine touch. A coated sample is securely double-sided taped to a stage, connected to two red sensors to record tangential and normal forces. A motion tracker is also placed on the moving finger to record displacement while sliding. Two setups are used simultaneously, one for each hand. (**B**) Formation of all instability phases during human exploration. (**C**) Tangential forces experienced by two hands on the same surface chemistry.

exponent seems to work for accuracy in the surfaces here, it may not be generalizable to other types of surfaces.

## Confirming instability formation during human exploration

To confirm that humans indeed form similar instabilities as the mock finger during exploration, we measured the tangential and normal force during a two-alternative forced choice (2-AFC) task. The participant placed their hands on a sensor stage, with one sample of **APTMS** under one hand and one sample of **C4** under the other. (*Figure 3A*) The participant was then asked to explore each sample simultaneously and ran over each surface in strokes along a single axis until the participant could decide which of the two had 'more friction'. *Figure 3B–C* shows forces generated during exploration. As expected, the participant had higher variability in their force trace than a mock finger. Yet despite this variability, we see that friction instabilities identified in the mock finger are also present in the human finger. We observe characteristic formation of steady sliding, slow frictional waves, and stiction spikes. Slow frictional waves are notable because people are unlikely to intentionally oscillate their finger in a rapid and consistent manner, especially since they were given no instruction to intentionally form the oscillations and generally slid their finger in a single motion.

## Conclusions

Here, participants freely explored 'tactile minimal' surfaces, which would not be expected to feel different to people based on nearly any common metric used in tactile studies. We further removed any confounding factors by asking participants to perform a low-level discrimination task, that is 'which one does not feel like the others?', instead of a subjective percept, like 'which one feels rougher?'. These human discrimination accuracy and response time were then correlated to surface characterization based on a mock finger. Despite apparent differences between a mock finger and a human finger, we still found that human results correlated to surface characterization based on the mock finger. More specifically, surfaces that generated more stiction spikes on the mock finger correlated to higher accuracy in human results. Surfaces that generated more stiction spikes on a mock finger correlated to a faster decision by human participants. However, if we analyze human results based on the friction coefficients generated by a mock finger, we find a spurious negative correlation, which is spurious because it suggests that identical surfaces would be easier for people to distinguish than two distinct surfaces.

Many studies on tactile perception correlate human results with direct biomechanical measurements on a human finger. Instead, we used a mock finger to characterize surfaces and correlated human results to data generated by a mock finger. Therefore, despite observing mechanical instabilities arise from touch of a human finger, a limitation of our approach is that we did not directly relate mechanical instabilities that arose from human fine touch with their ability to discriminate surfaces. However, unlike biomechanical measurements, the mock finger system can be replicated by other researchers to create a common repository and quantification of samples for tactile studies. Second, as exhibited by human measurements in *Figure 3*, it is not yet known how humans integrate tactile evidence in tactile decision making, but the instabilities identified in this study may offer a route forward for a consistent and traceable classifier for 'tactile evidence' to develop tactile decision-making models in future work. This may overcome the challenge that the raw friction force and derived friction coefficients for a given experiment will vary with almost any input (*Adams et al., 2013*), especially in macroscopic friction testing. As expected in soft macroscopic friction, we saw large interquartile ranges in friction coefficients in a mock finger – the variability in friction coefficients on a single surface consistently approaches the differences between two surfaces. This variability would be expected to be even higher in real human fingers. Despite the correlative nature of this study, we still obtained high correlations compared to existing biomechanical studies (*Skedung et al., 2020*; *Willemet et al., 2021*; *Gueorguiev et al., 2016*), which we speculate is because instabilities are an important predictive phenomenon for models of human touch. We believe that biomechanical studies, including more sophisticated techniques, like spatially resolved force maps from digital image correlation (*Waters et al., 2020*), may yield stronger correlations and results if they analyze data based on instabilities.

Instabilities are generated in a human finger, and the transitions between phases in different surfaces bridge the gap between the length scales relevant to molecular and mechanical phenomena. Instabilities may also help explain some aspects of tactile constancy, or the ability to retain object recognition properties despite varying environmental or human factors (*Yoshioka et al., 2011*; *Fehlberg et al., 2024*), because instabilities themselves can be mathematically invariant to mass or velocity under certain conditions. Although the data gathered here came from a comparatively simple, single force sensor, we were still able to identify a trend that held across multiple different samples, which has been challenging for the field. For those designing tactile interfaces in haptics or consumer goods, or extracting features in tactile arrays and soft robotics, analyzing existing force data along instabilities may be more predictive than traditional material properties and more consistent than raw friction traces.

## Acknowledgements

We acknowledge support from the National Eye Institute of the NIH (R01EY032584-03). XPS was conducted at the Surface Analysis Facility at the University of Delaware (NSF CHE-1428149). Atomic force microscopy was conducted at the Delaware Biotechnology Institute's Bio-Imaging Center at the University of Delaware supported by grants from the NIH-NIGMS (P20 GM103446), the NSF (IIA-1301765), and the State of Delaware. JGAC is supported by NSF CAREER award 2234748. This study was conducted and approved by the Institutional Review Board of the University of Delaware (Project #1484385-7), with data from 10 healthy participants between the ages of 21 and 32.

## Additional information

### Funding

| Funder | Grant reference number | Author |
|---|---|---|
| National Institutes of Health | R01EY032584-03 | Maryanne Derkaloustian Pushpita Bhattacharyya Truc T Ngo Jared Medina Charles B Dhong |
| National Science Foundation | CHE-1428149 | Maryanne Derkaloustian |
| National Institutes of Health | P20 GM103446 | Maryanne Derkaloustian |
| State of Delaware | | Maryanne Derkaloustian |
| National Science Foundation | IIA-1301765 | Maryanne Derkaloustian |
| National Science Foundation | 2234748 | Josh GA Cashaback |

The funders had no role in study design, data collection and interpretation, or the decision to submit the work for publication.

### Author contributions

Maryanne Derkaloustian, Conceptualization, Formal analysis, Investigation, Methodology, Writing – original draft, Writing - review and editing; Pushpita Bhattacharyya, Josh GA Cashaback, Formal analysis, Methodology; Truc T Ngo, Jared Medina, Methodology; Charles B Dhong, Supervision, Investigation, Writing – original draft

### Author ORCIDs

Maryanne Derkaloustian ⓘ https://orcid.org/0000-0002-8971-2594
Josh GA Cashaback ⓘ https://orcid.org/0000-0002-8642-6648
Charles B Dhong ⓘ https://orcid.org/0000-0003-3568-9859

### Ethics

This study was conducted and approved by the Institutional Review Board of the University of Delaware (Project #1484385-7), with data from 10 healthy participants between the ages of 21 and 32. All participants provided informed consent.

Reviewer #2 (Public review): https://doi.org/10.7554/eLife.104543.3.sa1
Reviewer #3 (Public review): https://doi.org/10.7554/eLife.104543.3.sa2
Reviewer #4 (Public review): https://doi.org/10.7554/eLife.104543.3.sa3
Author response https://doi.org/10.7554/eLife.104543.3.sa4

## Additional files

### Supplementary files

MDAR checklist

Source data 1. Mechanical friction data.

### Data availability

Friction traces from mechanical testing data to determine classifications are publicly available in *Source data 1*.

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

# Appendix 1

## Atomic force microscopy

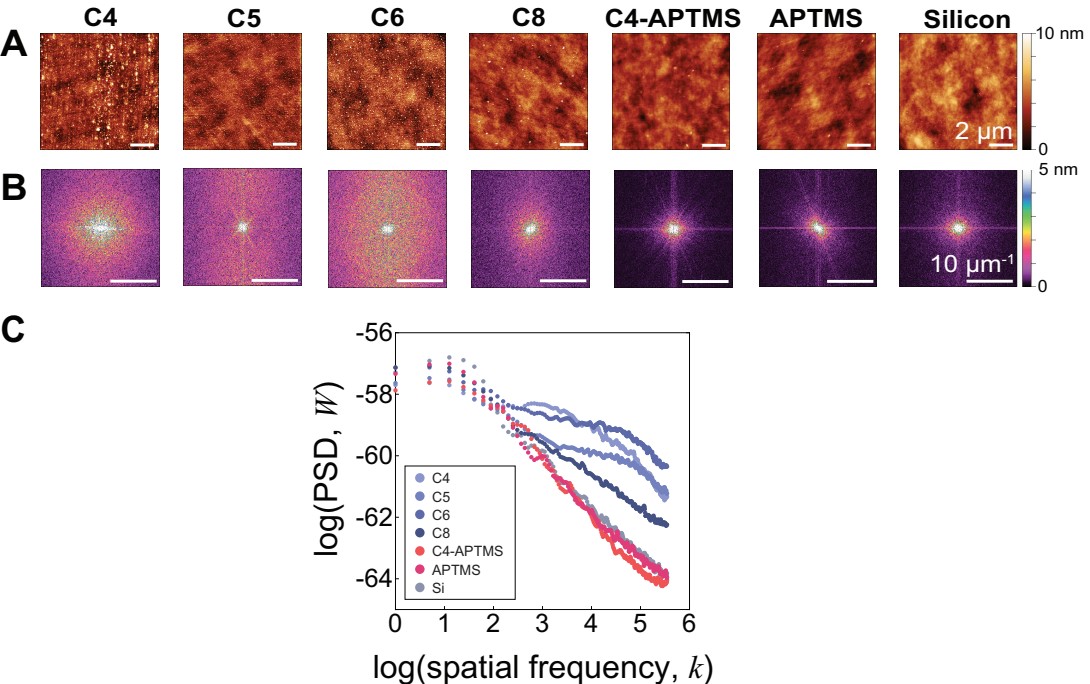

**Appendix 1—figure 1.** Surface characterization via atomic force microscopy. (**A**) Height profiles of surfaces obtained via AFM in tapping mode. All scale bars are 2 µm. (**B**) Respective 2D fast Fourier transforms of surface topographies. All scale bars are 10 µm⁻¹. (**C**) 2D power spectral densities of all surface profiles, plotted as log(PSD) vs. log(spatial frequency) to calculate Hurst exponents.

Average roughness ($R_a$) of the AFM topographies in ***Appendix 1—figure 1A*** was calculated along their diagonals (~14.14 µm). The values in ***Table 1*** are all on the small length scale of 100 pm, although **C4** and **C6** are slightly higher end due to the formation of more aggregates. This trend does not hold with **C8**; however, as the longer chains more easily form ordered films with less drastic height changes overall (***Xiao et al., 1996***). All alkylsilanes form a rougher coating than the aminosilanes, also consistent with literature (***Nolin et al., 2021***).

The two-dimensional fast Fourier transforms (2D FFTs) of the height profiles shown in ***Appendix 1— figure 1B*** are largely uniform. The shorter chain alkylsilanes present more diffuse rings in Fourier space, indicative of a lack of self-similarity across length scales (***Chrostowski et al., 2022***). Ordering and more fractal behavior begin to emerge with **C8**, as expected of longer chains with increased intermolecular interactions (***Nolin et al., 2021***; ***Xiao et al., 1996***). Meanwhile, the aminosilanes most closely replicate the underlying Si substrate in their isotropic, fractal behavior.

2D power spectral densities (PSDs) of the silanes were plotted in ***Appendix 1—figure 1C*** to quantify their scaling behavior through the Hurst exponent, *H*. This exponent identifies how roughness evolves across length scales: a lower *H* between 0 and 0.5 corresponds to a surface with a more homogeneous distribution of continuous high and low features, while *H* between 0.5 and 1 indicates the changes in roughness are sharper (***Nolin et al., 2021***; ***Ghosh and Pandey, 2019***). This is calculated as $H = 0.5 \times (|slope| - 1)$, using the slope of each surface's linear regime. A broad linear regime of smaller slope is observed for bare Si, as well as **C8**, **C4-APTMS**, and **APTMS**, resulting in lower *H* (see ***Table 1***). This is consistent with previous works, as van der Waals forces from increased chain length and hydrogen bonding from the amine groups both allow for chain alignment, with the roughness profile of uncoated Si well maintained and ordering preserved across most length scales (***Nolin et al., 2021***). This is in contrast to the profiles observed for **C4**, **C5**, and **C6**: multiple regimes appear, with steeper slopes observed at higher spatial frequencies. **C5** and **C6** still exhibit *H*<0.5, but are self-affine to a lower extent than **C8** and the aminosilane surfaces. **C4** demonstrates more

abrupt changes in height, even excluding the effects of the larger polymer aggregates ($H$=0.55 with masking). These higher $H$ thus represent increased disorder, including gauche defects and horizontal chain alignment (*Lio et al., 1997*), masking the underlying topography of bare Si.

## X-ray photoelectric spectroscopy

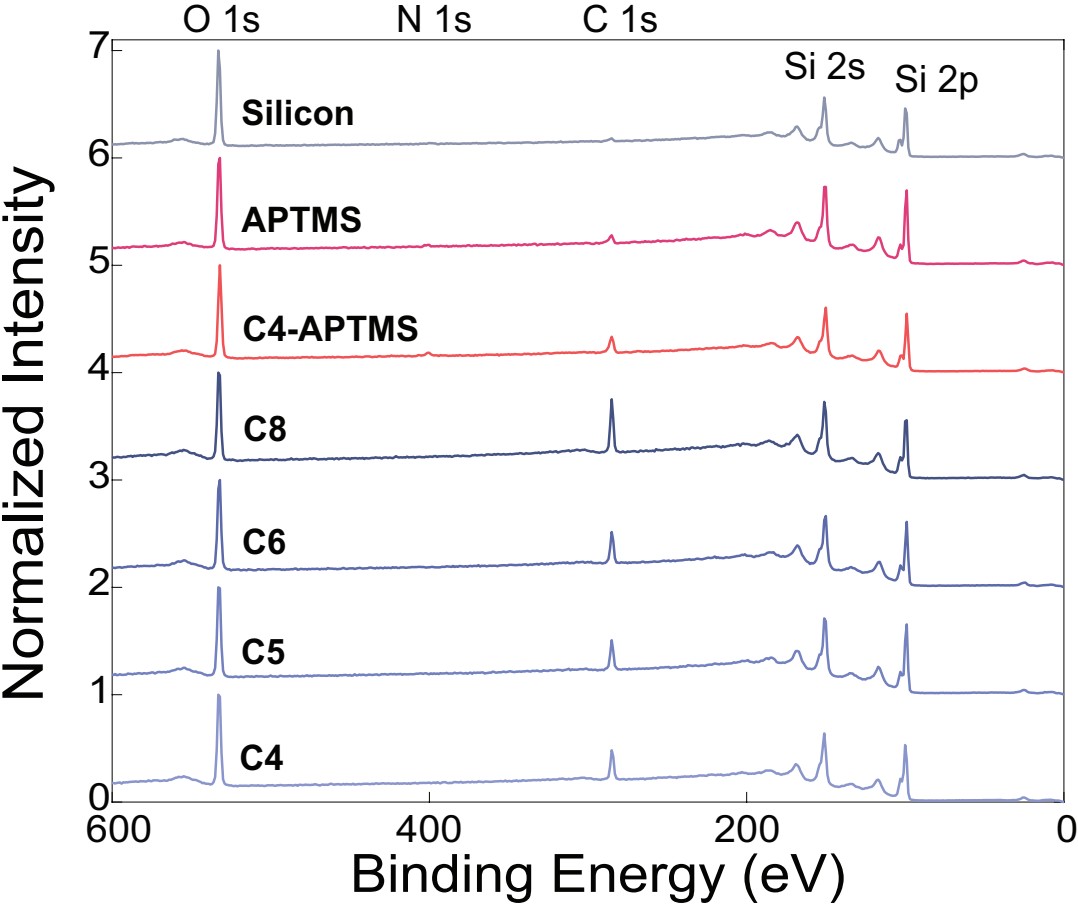

**Appendix 1—figure 2.** XPS survey scans of surfaces. C, N, and O peaks indicate evidence of binding onto Si wafers after deposition. Plotted as intensities normalized by the O peaks vs. binding energy.

XPS survey scans in *Appendix 1—figure 2* identify the elements present on our surfaces with a quantifiable photoelectric signal. To compare surfaces, all spectra were normalized by the highest peak with no expected change, corresponding to oxygen with a 532 eV peak (*Nagatomi et al., 1999*). Silanization is evident through the appearance of a carbon (285 eV) peak (*Wasserman et al., 1989*), of heights incrementally increasing with chain length in the alkylsilane surfaces (**C4**: 0.48, **C5**: 0.51, **C6**: 0.52, and **C8**: 0.75 × the height of the O peak). The C peaks are smaller but still visible in aminosilanes (**C4-APTMS**: 0.33 and **APTMS**: 0.30 × O). Small nitrogen peaks at 400 eV (0.19 × O) also appear in these two surfaces (*Liu et al., 2013*). Most importantly, the spectra present evidence of covalent bonding to the surfaces and successful deposition instead of physisorption. In the unreacted state, the alkylsilanes are chlorine-terminated, but no chlorine peak at 200 eV is present in the final surfaces (*Wasserman et al., 1989*; *Zhou and Leung, 2006*). Unreacted aminosilanes end with a methoxy group, but the C peaks from their deposited state are still small.

## Water contact angle hysteresis

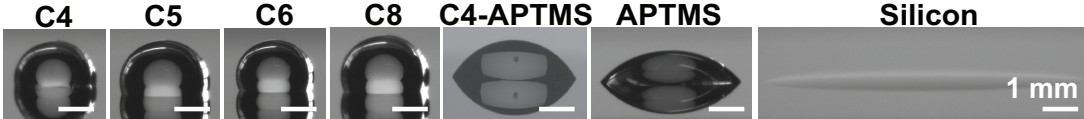

**Appendix 1—figure 3.** DI water droplets on surfaces, representative images used to measure advancing angles. All scale bars are 1 mm.

Even with differences in the AFM results, all the surfaces exhibit low chemical heterogeneity (*Nolin et al., 2021*; *Gao and McCarthy, 2006*), with water contact angle hysteresis <10°. There are noticeable differences in hydrophobicity, shown in the images used to measure advancing angles in *Appendix 1—figure 3*. The alkylsilanes form considerably rounder droplets indicating a higher degree of hydrophobicity than the aminosilanes (*Zhu et al., 2012*). However, even the aminosilanes are still more hydrophobic than the plasma-treated, uncoated Si, indicating the availability of free O allowing silanes to react and bind to the surfaces when undergoing depositions (*Suni et al., 2002*).

## Instability classification

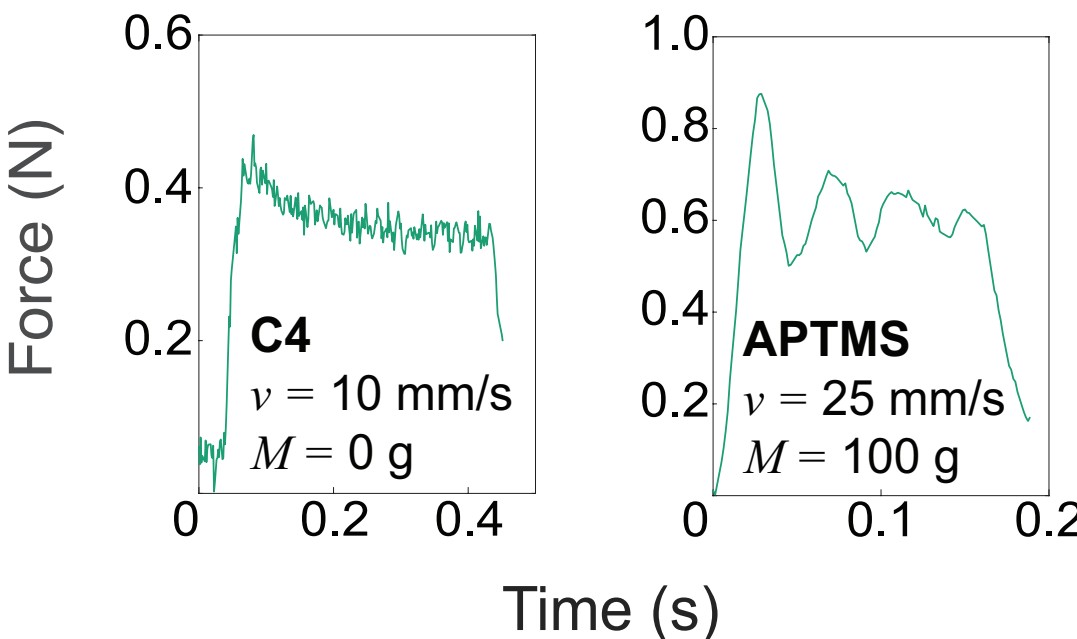

**Appendix 1—figure 4.** Force traces with less distinct categorization. Left: stiction spike with small height, right: stiction spike followed by small oscillations.

In many cases, instability classification was straightforward. However, in cases where stiction spikes were not as prominent as those in *Figure 1B*, only those at least 10% above the mean subsequent steady sliding were classified as stiction spikes. However, when the remainder of the trace still oscillates, a stiction spike was counted only if it was 40% higher in magnitude than the mean of the traces. Examples of these visually categorized stiction spikes are shown in *Appendix 1—figure 4*. Most importantly, these in-between cases mostly exist at boundary zones on our phase maps, where the same mass-velocity combinations do not produce identical force profiles. The boundaries do not only represent sharp transitions between instability phases, but also 'mixed-case' force traces.

## Statistical model selection

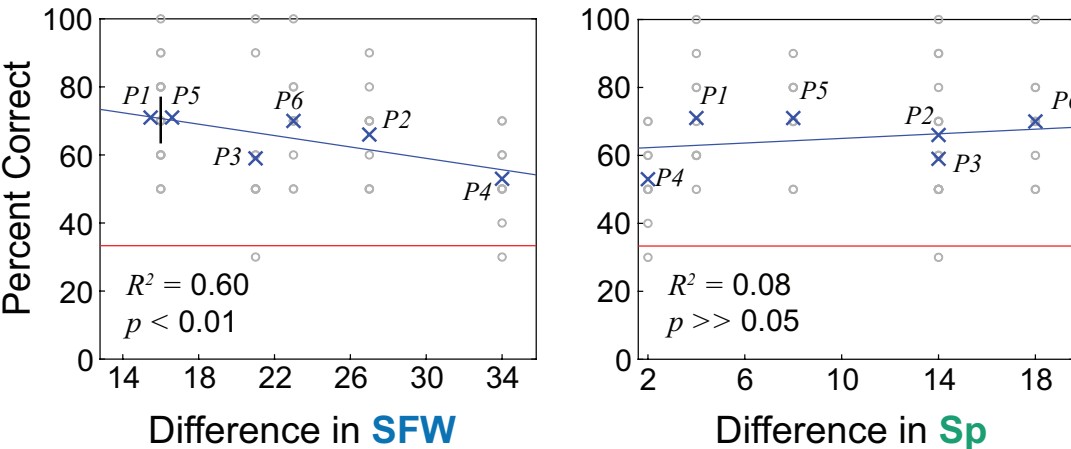

**Appendix 1—figure 5.** GLMM fits of participant accuracy vs. the differences in instability incidence for individual instability types. Left: accuracy vs. differences in formation of slow frictional waves (SFW) between pairs. *P1* and *P5* have the same *x*-axis value and are shifted for clarity. Right: accuracy vs. differences in formation of stiction spikes (Sp).

Participant data was fit to generalized linear mixed models (GLMMs) in order to decouple the variability of fixed and random effects, and for overall higher statistical power (*Moscatelli et al., 2012*). We first assessed subject accuracy by fitting normally distributed accuracy to differences in each instability type between pairs, as we hypothesized that all three instability types were important in predicting human performance. When examining each type individually, we first observed a positive, statistically significant relationship between accuracy and steady sliding (p<0.05). We also saw a negative correlation between accuracy and slow frictional waves that was statistically significant (p<0.01), and no correlation between accuracy and stiction spikes (*Appendix 1—figure 5*). We attribute these findings to the locations of these instability zones on the material phase maps. However, fitting all instability differences as terms in one GLMM did not yield any statistically significant results. A similar approach was used to correlate response times to instability types, which showed a statistically significant, negative correlation between response time and stiction spikes (details in **main text**).

## Effects of material properties

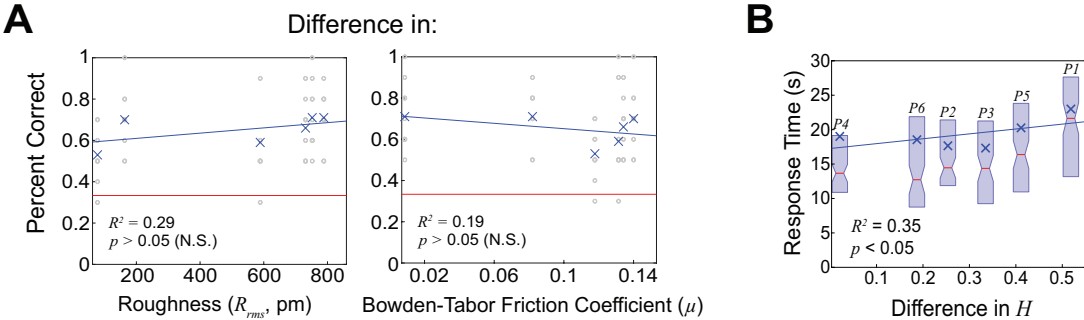

**Appendix 1—figure 6.** GLMMs of accuracy vs. more material parameters or properties across N=10 participants. (**A**) Left: root-mean-square roughness $R_{rms}$, right: velocity-dependent Bowden-Tabor friction coefficient across all conditions ($\mu$). (**B**) GLMM fit of response times vs. differences in Hurst exponent $H$. Mean times are represented by blue × marks, while medians are represented by red lines at notches of box plots.

After determining that $R_a$ was not a strong predictor, we also performed a GLMM fit on root-mean-square roughness $R_{rms}$, to verify that any common definition of roughness does not explain human performance. Although $R^2$ was slightly improved for this measure of roughness, each pair was

located similarly to $R_a$, leading to a fit that was still not statistically significant (***Appendix 1—figure 6A***).

The Bowden-Tabor friction coefficient of each material was determined by plotting average friction force vs. normal force for each velocity. Normal force was approximated as the applied load +the deadweight of the mock finger (6 g). Based on the equation $F = \mu N + \sigma A$, the slope of the points determined velocity-dependent friction coefficients (***Israelachvili, 2011***), which were then averaged to obtain a singular average friction coefficient for each material. Interestingly, these led to a negative fit similar to the more simplified friction coefficient but are not statistically significant.

As a positive correlation between accuracy and Hurst exponent was observed (p<0.05, details in main text), we fit the mean response times of each pair similarly. Here, we saw a positive, statistically significant relationship again (p<0.05, ***Appendix 1—figure 6B***), meaning the differences between pairs slow participants down. Unlike stiction spikes and frictional instabilities more broadly, each material's Hurst exponent is static, and participants cannot modulate their exploration conditions to feel other $H$ on the same material. No transitions exist to trigger a response, although after increased time, participants do successfully distinguish between $H$ on the silane surfaces.

