## [Editor Report · eLife Assessment]

This **useful** study integrates experimental methods from materials science with psychophysical methods to investigate how frictional stabilities influence tactile surface discrimination. The authors argue that force fluctuations arising from transitions between frictional sliding conditions facilitate the discrimination of surfaces with similar friction coefficients. However, the reliance on friction data from an artificial finger, combined with correlational analyses that fall short of establishing a mechanistic link to perception, renders the findings **incomplete**.

---

## [Referee Report · Reviewer #2 (Public review)]

This is a revised version of a paper I reviewed previously.

Again, the purpose of the paper is to suggest that common metrics, such as friction or any given physical property of the surface, are probably inadequate to predict the perception of the surface or its discriminability. Instead, the authors propose a very interesting and original idea that, instead, frictional instabilities are related to fine touch perception (title).

Overall, the authors have put much effort into improving the manuscript, enhancing clarity, and avoiding overstatements. And I feel the narrative is indeed much improved and less ambiguous.

However, the authors have systematically avoided addressing the main comment of all reviewers: the link made between the mock finger passive experiment and the active human psychophysics is incorrect and should not be done, because its interpretation could be flawed.

- First, this link is very weak (the correlation of 6 datapoints is barely significant).

- Second, the real and mock fingers have very different properties (think about moisture, compliance, roughness,...).

- Third, the comparison is made between a passive and well-controlled experiment and an active exploration. Yet, the comparison metrics (number of events) are clearly dependent on exploration procedures.

In your response to my comments:

"We have made changes throughout the manuscript to acknowledge that our findings are correlative, clarifying this throughout, and incorporating into the discussion how our work may enable biomechanical measurements and tactile decision making models"

The authors admit that the analysis is flawed, yet they did not remove it. If they cannot demonstrate that the mock finger and the human finger behave the same way during the perceptual experiment, then they should remove Fig2 that combines apples and oranges. OR, they should look at the active exploration data and compute the same metrics on that data.

"This "weird choice" is the central innovation of this paper. This choice was necessary because we demonstrated that the common usage of friction coefficient is fundamentally flawed: we see that friction coefficient suggests that surface which are more different would feel more similar - indeed the most distinctive surfaces would be two surfaces that are identical, which is clearly spurious. "

They did not "demonstrate" such a flaw. Again, the difference in friction is between the mock finger trials. At the very least, the authors should verify that it is true of the active human experiment.

"To fully implement this, a decision-making model is necessary because, as a counter example, a participant could have generated 10 swipes of SFW and 1 swipe of a Sp, but the Sp may have been the most important event for making a tactile decision. This type of scenario is not compatible with the analysis suggested - and similar counterpoints can be made for other types of seemingly straightforward analysis."

The suggested analyses are straightforward and would be much more valuable than the data from the mock finger, even with the potential variability stated above.

"We recognize that, with all factors being equal, this sample size is on the smaller end"

Yet, the authors did not collect additional data to confirm their findings.

---

## [Referee Report · Reviewer #3 (Public review)]

Strengths:

The paper describes a new perspective on friction perception, with the hypothesis that humans are sensitive to the instabilities of the surface rather than the coefficient of friction. The paper is very well written and with a comprehensive literature survey.

One of the central tools used by the author to characterize the frictional behavior is the frictional instabilities maps. With these maps, it becomes clear that two different surfaces can have both similar and different behavior depending on the normal force and the speed of exploration. It puts forward that friction is a complicated phenomenon, especially for soft

The psychophysics study is centered around an odd-one-out protocol, which has the advantage of avoiding any external reference to what would mean friction or texture for example. The comparisons are made only based on the texture being similar or not.

The results show a significant relationship between the distance between frictional maps and the success rate in discriminating two kinds of surface.

Weaknesses:

The main weakness of the paper comes from the fact that the frictional maps and the extensive psychophysics study are not made at the same time, nor with the same finger. The frictional maps are produced with an artificial finger made out of PDMS which is a poor substitute for the complex tribological properties of skin.

The evidence would have been much stronger if the measurement of the interaction was done during the psychophysical experiment. In addition, because of the protocol, the correlation is based on aggregates rather than on individual interactions. However the current data already bring new light on the nature of frictional oscillation and their link to perception.

The authors compensate with a third experiment where they used a 2AFC protocol and an online force measurement. But the results of this third study fail to solidify the relation.

No map of the real finger interaction is shown, bringing doubt to the validity of the frictional map for something as variable as human fingers.

---

## [Referee Report · Reviewer #4 (Public review)]

Summary:

In this paper, Derkaloustian et al. look at the important topic of what affects fine touch perception. The observations that there may be some level of correlation with instabilities are intriguing. They attempted to characterize different materials by counting the frequency (occurrence #, not of vibration) of instabilities at various speeds and forces of a PDMS slab pulled lengthwise over the material. They then had humans perform the same vertical motion to discriminate between these samples. They correlated the % correct in discrimination with differences in frequency of steady sliding over the design space as well as other traditional parameters such as friction coefficient and roughness.

The authors pose an interesting hypothesis and make an interesting observation about the occurrences of instability regimes in different materials while in contact with PDMS, which is interesting for the community to see in publication. It should be noted however that the finger is complex, and there are many factors that may be over simplified, and perhaps even incorrect, with the use of the PDMS finger. There are trends, such as the trend of surfaces that are more similar in PDMS friction coefficient being easier to discriminate than those with more different PDMS friction coefficient, that contradict multiple other papers in the literature (Fehlberg et al., 2024; Smith and Scott, 1996). This may be due to the PDMS finger not being representative of the real finger conditions. A measurement of friction and the instabilities with a human finger, or demonstration that the PDMS finger is producing the same results (friction coefficient, instabilities, etc.) as a human finger, is needed.

Strengths:

The strength of this paper is in its intriguing hypothesis and important observation that instabilities may contribute to what humans are detecting as differences in these apparently similar samples.

Weaknesses:

There is are significant weaknesses in the representativeness of the PDMS finger, the vertical motion, and the speed of sliding to real human exploration. The real finger has multiple layers with different moduli. In fact, the stratum corneum cells, which are the outer layer at the interface and determine the friction, have much higher modulus than PDMS. In addition, the flat contact area can cause shifting of contact points. Both can contribute to making the PDMS finger have much more stick slip than a real finger. In fact, if you look at the regime maps, there is very little space that has steady sliding. This does not represent well human exploration of surfaces. We do not tend to use force and velocity that will cause extensive stick slip (frequent regions of 100% stick slip) and, in fact, the speeds used in the study are on the slow side, which also contributes to more stick slip. At higher speeds and lower forces, all of the materials had steady sliding regions. Further, on these very smooth surfaces, the friction and stiction are more complex and cannot dismiss considerations such as finger material property change with sweat pore occlusion and sweat capillary forces. Also, the vertical motion of both the PDMS finger and the instructed human subjects is not the motion that humans typically use to discriminate between surfaces.

This all leads to the critical question, why is the friction, normal force, and velocity not measured during the measured human exploration using the real human finger? An alternative would be showing that the PDMS finger reproduces the results of the human finger. I have checked the author's previous papers with this setup and did not find one that showed that the PDMS finger produced the same results as a human finger (Carpenter et al., 2018; Dhong et al., 2018; Nolin et al., 2022, 2021). The reviewer is not asking to do a more detailed psychophysical study with a decision-making model. All that is being asked is to use a human finger for the friction coefficient and instability measurements at typical human forces and speeds, or at least doing these measurements with both for one or two samples to show that the PDMS finger produces the same results as a human finger. The authors posed an extremely interesting hypothesis that humans may alter their speed to feel the instability transition regions. This is something that could be measured with a real finger but is not likely to be correlated accurately enough to match regime boundaries determined with such a simplified artificial finger.

References

Carpenter CW, Dhong C, Root NB, Rodriquez D, Abdo EE, Skelil K, Alkhadra MA, Ramírez J, Ramachandran VS, Lipomi DJ. 2018. Human ability to discriminate surface chemistry by touch. Mater Horiz 5:70-77. doi:10.1039/C7MH00800G

Dhong C, Kayser LV, Arroyo R, Shin A, Finn M, Kleinschmidt AT, Lipomi DJ. 2018. Role of fingerprint-inspired relief structures in elastomeric slabs for detecting frictional differences arising from surface monolayers. Soft Matter 14:7483-7491. doi:10.1039/C8SM01233D

Fehlberg M, Monfort E, Saikumar S, Drewing K, Bennewitz R. 2024. Perceptual Constancy in the Speed Dependence of Friction During Active Tactile Exploration. IEEE Transactions on Haptics 17:957-963. doi:10.1109/TOH.2024.3493421

Nolin A, Licht A, Pierson K, Lo C-Y, Kayser LV, Dhong C. 2021. Predicting human touch sensitivity to single atom substitutions in surface monolayers for molecular control in tactile interfaces. Soft Matter 17:5050-5060. doi:10.1039/D1SM00451D

Nolin A, Pierson K, Hlibok R, Lo C-Y, Kayser LV, Dhong C. 2022. Controlling fine touch sensations with polymer tacticity and crystallinity. Soft Matter 18:3928-3940. doi:10.1039/D2SM00264G

Smith AM, Scott SH. 1996. Subjective scaling of smooth surface friction. Journal of Neurophysiology 75:1957-1962. doi:10.1152/jn.1996.75.5.1957

---

## [Author Response]

The following is the authors’ response to the original reviews

**eLife Assessment**
This useful study integrates experimental methods from materials science with psychophysical methods to investigate how frictional stabilities influence tactile surface discrimination. The authors argue that force fluctuations arising from transitions between frictional sliding conditions facilitate the discrimination of surfaces with similar friction coefficients. However, the reliance on friction data obtained from an artificial finger, together with the ambiguous correlative analyses relating these measurements to human psychophysics, renders the findings incomplete.

Our main goal with this paper was to show that the most common metric, i.e. average friction coefficient—widely used in tactile perception and device design – is fundamentally unsound, and to offer a secondary parameter that is compatible with the fact that human motion is unconstrained, leading to dynamic interfacial mechanics.

We understand the Reviewers wanted, through biomechanical measurements, to demonstrate that humans using instabilities. This is seemingly reasonable, but in individual responses, we explain the significant challenges and fundamental unknowns to those experiments. We believe this paper sets forth an important step to approach this problem. At the same time, we have made several changes in the discussion, conclusion, and title to clarify that our study is correlative between mechanical characterization and human testing.

In short, there are still several fundamental unknowns that prevented us from basing the study around biomechanical measurements: (1) a decision-making model would need to be created, but it is unknown if tactile decision making follows other models, (2) it is further unknown what constitutes “tactile evidence”, though at our manuscript’s conclusion, we propose that friction instabilities are better suited for to be tactile evidence than the averaging of friction coefficients from a narrow range of human exploration (3) in the design of samples, from a friction mechanics and materials perspective, it is not at this point, possible to pre-program surfaces *a priori* to deliver friction instabilities and instead must be experimentally determined – especially when attempting to achieve this in controlled surfaces that do not create other overriding tactile cues, like macroscopic bumps or large differences in surface roughness. (4) Given that the basis for tactile percepts, like which object feels “rougher” or “smoother” is not sufficiently established, it is necessary to use a 3-alternative forced choice task which avoids asking objects along a preset perceptual dimension – a challenge recognized by Reviewer 3. However, this would bring in issues of memory in the decision-making model. (5) The prior points are compounded by the fact that, we believe, tactile exploration must be performed in an unconstrained manner, i.e., without an apparatus generating motion onto a stationary finger. Work by Liu et al. (IEEE ToH, 2024) showed that recreating friction obtained during free exploration onto a stationary finger was uninterpretable by the participants, hinting at the importance of efference copies.[1] We believe that many of the above-mentioned issues constitutes a significant advance in knowledge and would require discussion and dissemination with the community.

Our changes to the manuscript

Page 1 & SI Page 1, Title

“Alternatives to Friction Coefficient: Fine Touch Perception Correlates with Frictional Instabilities”

**Reviewer 1 (Public review):**
Summary:In this paper, Derkaloustian et. al look at the important topic of what affects fine touch perception. The observations that there may be some level of correlation with instabilities are intriguing. They attempted to characterize different materials by counting the frequency (occurrence #, not of vibration) of instabilities at various speeds and forces of a PDMS slab pulled lengthwise over the material. They then had humans make the same vertical motion to discriminate between these samples. They correlated the % correct in discrimination with differences in frequency of steady sliding over the design space as well as other traditional parameters such as friction coefficient and roughness. The authors pose an interesting hypothesis and make an interesting observation about the occurrences of instability regimes in different materials while in contact with PDMS, which is interesting for the community to see in the publication. It should be noted that the finger is complex, however, and there are many factors that may be quite oversimplified with the use of the PDMS finger, and the consideration and discounting of other parameters are not fully discussed in the main text or SI. Most importantly, however, the conclusions as stated do not align with the primary summary of the data in Figure 2.Strengths:The strength of this paper is in its intriguing hypothesis and important observation that instabilities may contribute to what humans are detecting as differences in these apparently similar samples.

We thank Reviewer 1 for their time on the manuscript, recognizing the approach we took, and offering constructive feedback. We believe that our conclusions, in fact, are supported by the primary summary of the data in Fig. 2 but we believe that our use of *R2* could have led to misinterpretation. The trend with friction coefficient and percent correct was indeed statistically significant but was spurious because the slope was negative. In the revision, we add clarifying comments throughout, change from *R2* to *r* as to highlight the negative trend, and adjust the figures to better focus on friction coefficient.

Finally, we added a new section to discuss the tradeoffs between using a real human finger versus a mock finger, and which situations may warrant the use of one or the other. In short, for our goal of characterizing surfaces to be used in tactile experiments, we believe a mock finger is more sustainable and practical than using real humans because human fingers are unique per participant, humans move their fingers at constantly changing pressures and velocities, and friction generated during free exploring human cannot be satisfactorily replicated by moving a sample onto a stationary finger. But, we do not disagree that for other types of experiments, characterizing a human participant directly may be more advantageous.

Weaknesses:Comment 1The most important weakness is that the findings do not support the statements of findings made in the abstract. Of specific note in this regard is the primary correlation in Figure 2B between SS (steady sliding) and percent correct discrimination. Of specific note in this regard is the primary correlation in Figure 2B between SS (steady sliding) and percent correct discrimination. While the statistical test shows significance (and is interesting!), the R-squared value is 0.38, while the R-squared value for the "Friction Coefficient vs. Percent Correct" plot has an R-squared of 0.6 and a p-value of < 0.01 (including Figure 2B). This suggests that the results do not support the claim in the abstract: "We found that participant accuracy in tactile discrimination was most strongly correlated with formations of steady sliding, and response times were negatively correlated with stiction spikes. Conversely, traditional metrics like surface roughness or average friction coefficient did not predict tactile discriminability."

We disagree that the trend with friction coefficient suggests the results do not support the claim because the correlation was found to be negative. However, we could have made the comparison more apparent and expanded on this point, given its novelty.

While the *R2* value corresponding to the “Friction Coefficient vs. Percent Correct” plot is notably higher, our results show that the slope is negative, which would be statistically spurious. This is because a negative correlation between percent correct (accuracy in discriminating surfaces) and difference in friction coefficient means that the more similar two surfaces are (by friction coefficient), the easier it would be for people to tell them apart. That is, it incorrectly concludes that two identical surfaces would be much easier to tell apart than two surfaces with greatly different friction coefficients.

This is counterintuitive to nearly all existing results, but we believe our samples were well-positioned to uncover this trend by minimizing variability, by controlling multiple physical parameters in the samples, and that the friction coefficient — typically calculated in the field as an average friction coefficient — ignores all the dynamic changes in forces present in elastic systems undergoing mesoscale friction, i.e., human touch, as seen in Fig. 1 in a mock finger and Fig. 3 in a real finger. By demonstrating this statistically spurious trend, we believe this strongly supports our premise that an alternative to friction coefficient is needed in the design of tactile psychophysics and haptic interfaces.

We believe that this could have been misinterpreted, so we took several steps to improve clarity, given the importance of this finding: we separated the panel on friction coefficient to its own panel, we changed from *R2* to *r* throughout, and we added clarifying text. We also added a small section focusing on this spurious trend.

Our changes to the manuscript

Page 1, Abstract

“In fact, the typical method of averaging friction coefficients led to a spurious correlation which erroneously suggests that distinct objects should feel identical and identical objects should feel distinct.”

Page 7

“As Fig. 1 was constructed from friction measurements, we can also calculate an average friction coefficient, µ, by averaging the friction coefficient obtained at each of the 16 combinations of masses and velocities (Table 1). This calculation is a standard approach in tactile studies for summarizing friction measurements, or in some cases, surfaces are never characterized at multiple masses and velocities. However, summarizing friction data in this manner has been considered as conceptually questionable by others from a mechanics perspective.[3] Fig. 1 shows that the type of instabilities and friction forces encountered on a single surface can vary widely depending on the conditions. As a result, large variations in the friction coefficient are expected, depending on the mass and velocity — even though measurements originate from the same surface. This variability in friction coefficient can be seen with the large interquartile range of friction coefficients, which shows that the variation in friction coefficient across a single surface is similar, or even larger, than the differences in average friction coefficient across two different surfaces. The observation that friction coefficients vary so widely on a single surface calls into question the approach of analyzing how humans may perceive two different objects based on their average friction coefficients.”

Page 9, Fig. 2 Caption

“(D) GLMM of accuracy vs. difference in average friction coefficient \begin{document}$(\Delta \bar{\mu})$\end{document}, showing a negative correlation. (E) GLMMs of accuracy vs. other commonly used material properties or parameters: ΔAverage roughness *Ra*, ΔHurst exponent *H*, and ΔWater contact angle hysteresis (º) (*N =* 10 participants_,_ *n* = 600 total trials).”

Page 9

“Considering all instabilities individually, we found that only steady sliding was a positive, statistically significant predictor. (*r* = 0.62, *p* < 0.05, shown in Fig. 2B).”

Page 10

“To compare the value of looking at frictional instabilities, we also performed GLMM fits on common approaches in the field, like a friction coefficient or material property typically used in tactile discrimination, shown in Fig. 2D-E. Interestingly, in Fig. 2D, we observed a spurious, negative correlation between friction coefficient (typically and often problematically simplified as \begin{document}$F=\bar{\mu} N$\end{document} across all tested conditions) and accuracy (*r* = -0.64, *p* < 0.01); that is, the more different the surfaces are by friction coefficient, the less people can tell them apart. This spurious correlation would be the opposite of intuition, and further calls into question the common practice of using friction coefficients in touch-related studies. Interestingly, this spurious correlation was also found by Gueorguiev et al.[21] The alternative, two-term model which includes adhesive contact area for friction coefficient[32] was even less predictive (see Fig. S6A of SI). We believe such a correlation could not have been uncovered previously as our samples are minimal in their physical variations. Yet, the dynamic changes in force even within a single sample are not considered, despite being a key feature of mesoscale friction during human touch.

We investigate different material properties in Fig. 2E. Differences in average roughness *Ra* (or other parameters, like root mean square roughness *Rrms* Fig. S6A of SI) did not show a statistically significant correlation to accuracy. Though roughness is a popular parameter, correlating any roughness parameter to human performance here could be moot: the limit of detecting roughness differences has previously been defined as 13 nm on structured surfaces[36] and much higher for randomly rough surfaces,[49] all of which are magnitudes larger than the roughness differences between our surfaces. The differences in contact angle hysteresis – as an approximation of the adhesion contributions[50] – do not present any statistically significant effects on performance.”

Page 11-12

“Despite the correlative nature of this study, we still obtained high correlations compared to existing biomechanical studies[4,19,21], which we speculate is because instabilities are an important predictive phenomenon for models of human touch. We believe that biomechanical studies, including more sophisticated techniques, like spatially resolved force maps from digital image correlation[5,42] may yield stronger correlations and results if they analyze data based on instabilities.

Added References

(2) Khamis, H. et al. Friction sensing mechanisms for perception and motor control: passive touch without sliding may not provide perceivable frictional information. J. Neurophysiol. 125, 809– 823 (2021).

(6) Olczak, D., Sukumar, V. & Pruszynski, J. A. Edge orientation perception during active touch. J. Neurophysiol. 120, 2423–2429 (2018).

Comment 2, Part 1Along the same lines, other parameters that were considered such as the "Percent Correct vs. Difference in Sp" and "Percent Correct vs. Difference in SFW" were not plotted for consideration in the SI. It would be helpful to compare these results with the other three metrics in order to fully understand the relationships.

We have added these plots to the SI. We note that we had checked these relationships and discussed them briefly, but did not include the plot. The plots show that the type of instability was not as helpful as its presence or absence.

Our changes to the manuscript

Page 9

“Furthermore, a model accounting for slow frictional waves alone specifically shows a significant, negative effect on performance (*p* < 0.01, Fig. S5 of SI), suggesting that in these samples and task, the type of instability was not as important.”

“Fig. S5. GLMM fits of participant accuracy vs. the differences in instability incidence for individual instability types. Left: accuracy vs. differences in formation of slow frictional waves (SFW) between pairs. *P1* and *P5* have the same *x*-axis value and are shifted for clarity. Right: accuracy vs. differences in formation of stiction spikes (Sp).”

SI Page 4

“and no correlation between accuracy and stiction spikes (Fig. S5).”

Comment 2, Part 2Other parameters such as stiction magnitude and differences in friction coefficient over the test space could also be important and interesting.

We agree these are interesting and have thought about them. We are aware that others, like Gueorguiev et al., have studied stiction magnitudes, and though there was a correlation, the physical differences in surface roughness (glass versus PMMA) investigated made it unclear if these could be generalized further.[3] We are unsure how to proceed here with a satisfactory analysis of stiction magnitude, given that stiction spikes are not always generated. In fact, Fig. 1 shows that for many velocities and pressures, stiction spikes are not formed. In ongoing work, however, we are always cognizant that if stiction spikes are a dominant factor, then a secondary analysis on their magnitude would be important. We offer some speculation on why stiction spikes may be overrepresented in the literature:

(1) They are prone to being created if the finger was loaded for a long time onto a surface prior to movement, thus creating adhesion by contact aging which is unlike active human exploration. We avoid this by discarding the first pull in our measurements, which is a standard practice in mechanical characterization if contact aging needs to be avoided.

(2) The ranges of velocities and pressures explored by others were small.

(3) In an effort to generate strong tactile stimuli, highly adhesive or rough surfaces are used.

(4) Stiction spikes are visually distinctive on a plot, but we are unaware of any mechanistic reason that mechanoreceptors would be particularly sensitive to this low frequency event over other signals.

We interpret “difference in friction coefficient over the test space” to be, for a single surface, like C4, to find the highest average friction for a condition of single velocity and mass and subtract that from the lowest average friction for a condition of single velocity and mass. We calculated the difference in friction coefficient in the typical manner of the field, by averaging all data collected at all velocities and masses and assigning a single value for all of a surface, like C4. We had performed this, and have the data, but we are wary of overinterpreting secondary and tertiary metrics because they do not have any fundamental basis in traditional tribology, and this value, if used by humans, would suggest that they rapidly explore a large parameter space to find a “maximum” and “minimum” friction. Furthermore, the range in friction across the test space, after averaging, can be smaller than the range of friction experienced at different masses and velocities on a single surface. We have tabulated and newly included these values (the interquartile range of friction coefficients of different masses and velocities per surface) in Table 1.

Fig. 2D shows a GLMM fit between percent correct responses across our pairs and the differences in friction coefficient for each pair, where we see a spurious negative correlation. As we had the data of all average friction coefficients for each condition for a given material, we also looked at the difference in maximum and minimum friction coefficients. For our tested pairs, these differences also lined up on a statistically significant, negative GLMM fit (*r* = -0.86, *p* < 0.005). However, the values for a given surface can vary drastically, with an interquartile range of 1.20 to 2.09 on a single surface. We fit participant accuracy to the differences in these IQRs across pairs. This also led to a negative GLMM fit (*r* = -0.65, *p* < 0.05). However, we are hesitant to add this plot to the manuscript for the reasons stated previously.

Comment 3, Part 1Beyond this fundamental concern, there is a weakness in the representativeness of the PDMS finger, the vertical motion, and the speed of sliding to real human exploration.

Overall, this is a continuous debate that we think offers two solutions, and we are not advocating for an “either-or” case. There is always a tradeoff between using a synthetic model of a finger versus a real human finger, and there is a place for both models. That is, while our mock finger will be “better” the more similar it is to a human finger, it is not our goal to fully replace a human finger. Rather our goal is to provide a consistent method of characterizing surfaces that is sufficiently similar to human touch as to be a useful and predictive tool.

The usefulness of the mock finger is in isolating the features of each surface that is independent of human variability, i.e., instabilities that form without changing loading conditions between sliding motions or even within one sliding motion. Of course, with this method, we still require confirmation of these features still forming during human exploration, which we show in Fig. 3. We believe that this method of characterizing surfaces at the mesoscale will ultimately lead to more successful human studies on tactile perception. Currently, and as shown in the paper, characterizing surfaces through traditional techniques, such as a commercial tribometer (friction coefficient, using a steel or hard metal ball), roughness (via atomic force microscopy or some other metrology), surface energy are less or not at all predictive. Thus, we believe this mock finger is better than the current state-of-the-art characterizing surfaces (we are also aware of a commercial mock finger company, but we were unable to purchase or obtain an evaluation model).

One of the main – and severe – limitations of using a human finger is that all fingers are different, meaning any study focusing on a particular user may not apply to others or be recreated easily by other researchers. We do not think it is feasible to set a standard for replication around a real human finger as that participant may no longer be available, or willing to travel the world as a “standard”. Furthermore, the method in which a person changes their pressures and velocities is different. We note that this is a challenge unique to touch perception – how an object is touched changes the friction generated, and thus the tactile stimulus generated, whereas a standardized stimulus is more straightforward for light or sound.

However, we do emphasize that we have strongly considered the balance between feasibility and ecological validity in the design of a mock finger. We have a mock finger, with the three components of stiffness of a human finger (more below). Furthermore, we have also successfully used this mock finger in correlations with human psychophysics in previous work, where findings from our mechanical experiments were more predictive of human performance[4–7] than other available methods.

Our changes to the manuscript Added (Page 2-3)

“Mock finger as a characterization tool

We use a mechanical setup with a PDMS (poly(dimethylsiloxane)) mock finger to derive tactile predictors as opposed to direct biomechanical measurements on human participants. While there is a tradeoff in selecting a synthetic finger over a real human finger to modeling human touch, human fingers themselves are also highly variable[23] both in their physical shape and their use during human motion. Our goal is to design a consistent method of characterization of samples that can be easily accessed by other researchers and does not rely on a standard established around single human participant. We believe that sufficient replication of surface, bulk properties, and contact geometry results in characterization that isolates consistent features of surfaces that are not derived from human-to-human variability. We have used this approach to successfully correlate human results with mock finger characterization previously.[8,9,24]

The major component of a human finger, by volume, is soft tissue (~56%),[25] resulting in an effective modulus close to 100 kPa.[26,27] In order to achieve this same softness, we crosslink PDMS in a 1×1×5 cm mold at a 30:1 elastomer:crosslinker ratio. In addition, two more features in the human finger impart significant mechanical differences. Human fingers have a bone at the fingertip, the distal phalanx,[26–28, 8–10]which we mimic with an acrylic “bone” within our PDMS network. The stratum corneum, the stiffer, glassier outer layer of skin,[29] is replicated with the surface of the mock finger glassified, or further crosslinked, after 8 hours of UV-Ozone treatment.30 This treatment also modifies the surface properties of the native PDMS to align with those of a human finger more closely: it minimizes the viscoelastic tack at the surface, resulting in a comparable non-sticky surface. Stabilizing after one day after treatment, the mock finger surface obtains a moderate hydrophilicity (~60º), as is typically observed for a real finger.[11,31]

The initial contact area formed before a friction trace is collected is a rectangle of 1×1 cm. While this shape is not entirely representative of a human finger with curves and ridges, human fingers flatten out enough to reduce the effects of curvature with even very light pressures.[31–33] This implies that for most realistic finger pressures, the contact area is largely load-independent, which is more accurately replicated with a rectangular mock finger.

Lastly, we consider the role of fingerprint ridges. A key finding of our previous work is that while fingerprints enhanced frictional dynamics at certain conditions, key features were still maintained with a flat finger.[11] Furthermore, for some loading conditions, the more amplified signals could also result in more similar friction traces for different surfaces. We have observed good agreement between these friction traces and human experiments.[8,9,22,34]”

Page 3-4, Materials and Methods

“Mock Finger Preparation

Friction forces across all six surfaces were measured using a custom apparatus with a polydimethylsiloxane (PDMS, Dow Sylgard 184) mock finger that mimics a human finger’s mechanical properties and contact mechanics while exploring a surface relatively closely.[8,9] PDMS and crosslinker were combined in a 30:1 ratio to achieve a stiffness of 100 kPa comparable to a real finger, then degassed in a vacuum desiccator for 30 minutes. We are aware that the manufacturer recommended crosslinking ratio for Sylgard 184 is 10:1 due to potential uncrosslinked liquid residues,[35] but further crosslinking concentrated at the surface prevents this. The prepared PDMS was then poured into a 1×1×5 cm mold also containing an acrylic 3D-printed “bone” to attach applied masses on top of the “fingertip” area contacting a surface during friction testing. After crosslinking in the mold at 60ºC for 1 hour, the finger was treated with UV-Ozone for 8 hours out of the mold to minimize viscoelastic tack.

Mechanical Testing

A custom device using our PDMS mock finger was used to collect macroscopic friction force traces replicating human exploration.[8,9] After placing a sample surface on a stage, the finger was lowered at a slight angle such that an initial 1×1 cm rectangle of “fingertip” contact area could be established. We considered a broad range of applied masses (*M* = 0, 25, 75, and 100 g) added onto the deadweight of the finger (6 g) observed during a tactile discrimination task. The other side of the sensor was connected to a motorized stage (V-508 PIMag Precision Linear Stage, Physikinstrumente) to control both displacement (4 mm across all conditions) and sliding velocity (*v* = 5, 10, 25, and 45 mm s^-1^). Forces were measured at all 16 combinations of mass and velocity via a 250 g Futek force sensor (*k* = 13.9 kN m^-1^) threaded to the bone, and recorded at an average sampling rate of 550 Hz with a Keithley 7510 DMM digitized multimeter. Force traces were collected in sets of 4 slides, discarding the first due to contact aging. Because some mass-velocity combinations were near the boundaries of instability phase transitions, not all force traces at these given conditions exhibited similar profiles. Thus, three sets were collected on fresh spots for each condition to observe enough occurrences of multiple instabilities, at a total of nine traces per combination for each surface.”

Added References

(23) Infante, V. H. P. et al. The role of skin hydration, skin deformability, and age in tactile friction and perception of materials. Sci. Rep. 15, 9935 (2025).

(24) Nolin, A., Lo, C.-Y., Kayser, L. V. & Dhong, C. B. Transparent and Electrically Switchable Thin Film Tactile Actuators Based on Molecular Orientation. Preprint at https://doi.org/10.48550/arXiv.2411.07968 (2024).

(25) Murai, M., Lau, H.-K., Pereira, B. P. & Pho, R. W. H. A cadaver study on volume and surface area of the fingertip. J. Hand Surg. 22, 935–941 (1997).

(26) Abdouni, A. et al. Biophysical properties of the human finger for touch comprehension: influences of ageing and gender. R. Soc. Open Sci. (2017) doi:10.1098/rsos.170321.

(27) Cornuault, P.-H., Carpentier, L., Bueno, M.-A., Cote, J.-M. & Monteil, G. Influence of physico-chemical, mechanical and morphological fingerpad properties on the frictional distinction of sticky/slippery surfaces. J. R. Soc. Interface (2015) doi:10.1098/rsif.2015.0495.

(28) Qian, K. et al. Mechanical properties vary for different regions of the finger extensor apparatus. J. Biomech. 47, 3094–3099 (2014).

(29) Yuan, Y. & Verma, R. Measuring microelastic properties of stratum corneum. Colloids Surf. B Biointerfaces 48, 6–12 (2006).

(30) Fu, Y.-J. et al. Effect of UV-Ozone Treatment on Poly(dimethylsiloxane) Membranes: Surface Characterization and Gas Separation Performance. Langmuir 26, 4392–4399 (2010).

Comment 3, Part 2The real finger has multiple layers with different moduli. In fact, the stratum corneum cells, which are the outer layer at the interface and determine the friction, have a much higher modulus than PDMS. The real finger has multiple layers with different moduli. In fact, the stratum corneum cells, which are the outer layer at the interface and determine the friction, have a much higher modulus than PDMS.

We have approximated the softness of the finger with 100 kPa crosslinked PDMS, which is close to what has been reported for the bulk of a human fingertip.[9,10] However, as mentioned in the Materials and Methods, there are two additional features of the mock finger that impart greater strength. The PDMS surrounds a rigid, acrylic bone comparable to the distal phalanx, which provides an additional layer of higher modulus.[8] Additionally, the 8-hour UV-Ozone treatment decreases the viscoelastic tack of the pristine PDMS by glassifying, or further crosslinking the surface of the finger,[12] therefore imparting greater stiffness at the surface similar to the contributions of the stratum corneum, along with a similar surface energy.[13] This technique is widely used in wearables,[14] soft robotics,[15] and microfluidics[16] to induce both these material changes. Additionally, the finger is used at least a day after UV-Ozone treatment is completed to generate a stable surface that is moderately hydrophilic, similar to the outermost layer of human skin.[17]

Comment 3, Part 3In addition, the slanted position of the finger can cause non-uniform pressures across the finger. Both can contribute to making the PDMS finger have much more stick-slip than a real finger.

To ensure that there is minimal contribution from the slanted position of the finger, an initial contact area of 1×1 cm is established before sliding and recording friction measurements. As the PDMS finger is a soft object, the portion in contact with a surface flattens and the contact area remains largely unchanged during sliding. Any additional stick-slip after this alignment step is caused by contact aging at the interface, but the first trace we collect is always discarded to only consider stick-slip events caused by surface chemistry. We recognize that it is difficult to completely control the pressure distribution due to the planar interface, but this is also expected when humans freely explore a surface.

Comment 3, Part 4In fact, if you look at the regime maps, there is very little space that has steady sliding. This does not represent well human exploration of surfaces. We do not tend to use a force and velocity that will cause extensive stick-slip (frequent regions of 100% stick-slip) and, in fact, the speeds used in the study are on the slow side, which also contributes to more stick-slip. At higher speeds and lower forces, all of the materials had steady sliding regions.”

We are not aware of published studies that extensively show that humans avoid stickslip regimes. In fact, we are aware familiar with literature where stiction spike formation is suppressed – a recent paper by AliAbbasi, Basdogan et. al. investigates electroadhesion and friction with NaCl solution-infused interfaces, resulting in significantly steadier forces.[18] We also directly showed evidence of instability formation that we observed during human exploration in Fig. 3B-C. These dynamic events are common, despite the lack of control of normal forces and sliding velocities. We also note that Reviewer 1, Comment 2, Part 2 was suggesting that we further explore possible trends from parameterizing the stiction spike.

We note that many studies have often not gone at the velocities and masses required for stiction spikes – even though these masses and velocities would be routinely seen in free exploration – this is usually due to constraints of their equipment.[19] Sliding events during human free exploration of surfaces can exceed 100 mm/s for rapid touches. However, for the surfaces investigated here, we observe that large regions of stick-slip can emerge at velocities as low as 5 mm/s depending on the applied load. The incidence of steady sliding appears more dependent on the applied mass, with almost no steady sliding observed at or above 75 g. Indeed, the force categorization along our transition zones is the main point of the paper.

Comment 3, Part 5Further, on these very smooth surfaces, the friction and stiction are more complex and cannot dismiss considerations such as finger material property change with sweat pore occlusion and sweat capillary forces. Also, the vertical motion of both the PDMS finger and the instructed human subjects is not the motion that humans typically use to discriminate between surfaces.

We did not describe the task sufficiently. Humans were only given the instruction to slide their finger along a single axis from top to bottom of a sample, not vertical as in azimuthal to gravity. We have updated our wording in the manuscript to reflect this.

Page 4

“Participants could touch for as long as they wanted, but were asked to only use their dominant index fingers along a single axis to better mimic the conditions for instability formation during mechanical testing with the mock finger.”

Page 11

“The participant was then asked to explore each sample simultaneously, and ran over each surface in strokes along a single axis until the participant could decide which of the two had “more friction”.”

Comment 3, Part 6Finally, fingerprints may not affect the shape and size of the contact area, but they certainly do affect the dynamic response and detection of vibrations.”

We are aware of the nuance. Our previous work on the role of fingerprints on friction experienced by a PDMS mock finger showed enhanced signals with the incorporation of ridges on the finger and used a rate-and-state model of a heterogenous, elastic body to find corresponding trends (though there is no existing model of friction that can accurately model experiments on mesoscale friction).[11] The key conclusion was that a flat finger still preserved key dynamic features, and the presence of stronger or more vibrations could result in more similar forces for different surfaces depending on the sliding conditions.

This is also in the context that we are seeking to provide a reasonable and experimentally accessible method to characterize surfaces, which will always be better as we get closer in replicating a true human finger. But our goal here was to replicate the finger sufficiently for use in human studies. We believe the more appropriate metric of success is if the mock finger is more successful than replacing traditional characterization experiments, like friction coefficient, roughness, surface energy, etc.

Comment 4This all leads to the critical question, why are friction, normal force, and velocity not measured during the measured human exploration and in a systematic study using the real human finger? The authors posed an extremely interesting hypothesis that humans may alter their speed to feel the instability transition regions. This is something that could be measured with a real finger but is not likely to be correlated accurately enough to match regime boundaries with such a simplified artificial finger.

We are excited that our manuscript offers a tractable manner to test the hypothesis that tactile decision-making models use friction instabilities as evidence. However, we lay out the challenges and barriers, and how the scope of this paper will lead us in that direction. We also clarify that our goals are to provide a method to characterize samples to better design tactile interfaces in haptics or in psychophysical experiments and raise awareness that the common methods of sample characterization in touch by an average friction coefficient or roughness is fundamentally unsound. Throughout the paper, we have made changes to reflect that our study, at this point, is only correlative.

As discussed in the summary, and with additional detail here, to further support our findings through observation on humans would require answering:

(1) Which one, or combination of, of the multiple swipes that people make responsible for a tactile decision? (There is a need for a decision-making model)

(2) Establish what is, or may be, tactile evidence.

(3) Establish tactile decision-making models are similar or different than existing decision-making models.

(4) Design a task that does not require the use of subjective tactile descriptors, like “which one feels rougher”, which we have seen causes confusion in participants, which will likely require accounting for memory effects.

We elaborate these points below:

To successfully perform this experiment, we note that freely exploring humans make multiple strokes on a surface. Therefore, we would need to construct a decision-making model. It has not yet been demonstrated whether tactile decision making follows visual decision making, but perhaps to start, we can assume it does. Then, in the design of our decision-making paradigm, we immediately run into the problem: What is tactile evidence?

From Fig. 3C, we already can see that identifying evidence is challenging. Prior to this manuscript, people may have chosen the average force, or the highest force. Or we may choose the average friction force. Then, after deciding on the evidence, we need to find a method to manipulate the evidence, i.e., create samples or a machine that causes high friction, etc. We show that during the course of human touch, due to the dynamic nature of friction, the average can change a large amount and sample design becomes a central barrier to experiments. Others may suggest immobilizing the finger and applying a known force, but given how much friction changes with human exploration, there is no known method to make a machine recreate temporally and spatially varying friction forces during sliding onto a stationary finger. Finally, perhaps most importantly, in addition to mechanical challenges, a study by Liu, Colgate et al. showed that even if they recorded the friction (2D) of a finger exploring a surface and then replicated the same friction forces onto a finger, the participant could not determine which surface the replayed friction force was supposed to represent.[1] This supports that the efference copy is important, that the forces in response to expected motion are important to determine friction. Finally, there is no known method to design instabilities *a priori*. They must be found through experiments. Especially since if we were to introduce, say a bump or a trough, then we bring in confounding variables to how participants tell surfaces apart.

Furthermore, even if we had some consistent method to create tactile “evidence”, the paradigm also deserves some consideration. In our experience, the 3-AFC task we perform is important because the vocabulary for touch has not been established. That is, in 3-AFC, by asking to determine which one sample is unlike the others, we do not have to ask the participant questions like “which one is rougher” or “which one has less friction”. In contrast, 2-AFC, which is better for decision-making models because it does not include memory, requires the asking of a perceptual question like: “which one is rougher?”. In our ongoing work, taking two silane coatings, we found that participants could easily identify which surface is unlike the others above chance in a 3-AFC, but participants, even within their own trials, could not consistently identify one silane as perceptually “rougher” by 2-AFC. To us, this calls into question the validity of tactile descriptors, but is beyond the scope of this manuscript.

This is not our only goal, but in the context of human exploration, in this manuscript here, we believed it was important to identify a mechanical parameter that was consistent with how humans explore surfaces, but was also a parameter that could characterize to some consistent property of a surface – irrespective of whether a human was touching it. We thought that designing human decision-making models and paradigms around the friction coefficient would not be successful.

Given the scope of these challenges, we do not think it would be possible to establish these conceptual sequences in a single manuscript. However, we think that our manuscript brings an important step forward to approach this problem.

**Reviewer 2 (Public review):**
Summary:In this paper, the authors want to test the hypothesis that frictional instabilities rather than friction are the main drivers for discriminating flat surfaces of different sub-nanometric roughness profiles.They first produced flat surfaces with 6 different coatings giving them unique and various properties in terms of roughness (picometer scale), contact angles (from hydrophilic to hydrophobic), friction coefficient (as measured against a mock finger), and Hurst exponent.Then, they used those surfaces in two different experiments. In the first experiment, they used a mock finger (PDMS of 100kPA molded into a fingertip shape) and slid it over the surfaces at different normal forces and speeds. They categorized the sliding behavior as steady sliding, sticking spikes, and slow frictional waves by visual inspection, and show that the surfaces have different behaviors depending on normal force and speed. In a second experiment, participants (10) were asked to discriminate pairs of those surfaces. It is found that each of those pairs could be reliably discriminated by most participants.Finally, the participant's discrimination performance is correlated with differences in the physical attributes observed against the mock finger. The authors found a positive correlation between participants' performances and differences in the count of steady sliding against the mock finger and a negative correlation between participants' reaction time and differences in the count of stiction spikes against the mock finger. They interpret those correlations as evidence that participants use those differences to discriminate the surfaces.Strengths:The created surfaces are very interesting as they are flat at the nanometer scale, yet have different physical attributes and can be reliably discriminated.

We thank Reviewer 2 for their notes on our manuscript. The responses below address the reviewer’s comments and recommendations for revised work.

Weaknesses:Comment 1In my opinion, the data presented in the paper do not support the conclusions. The conclusions are based on a correlation between results obtained on the mock finger and results obtained with human participants but there is no evidence that the human participants' fingertips will behave similarly to the mock finger during the experiment. Figure 3 gives a hint that the 3 sliding behaviors can be observed in a real finger, but does not prove that the human finger will behave as the mock finger, i.e., there is no evidence that the phase maps in Figure 1C are similar for human fingers and across different people that can have very different stiffness and moisture levels.

We have made changes throughout the manuscript to acknowledge that our findings are correlative, clarifying this throughout, and incorporating into the discussion how our work may enable biomechanical measurements and tactile decision making models.

The mechanical characterization conducted with the mock finger seeks to extract significant features of friction traces of a set of surfaces to use as predictors of tactile discriminability. The goal is to find a consistent method to characterize surfaces for use in tactile experiments that can be replicated by others and used prior to any human experiments. However, in the overall response and in a response to a similar comment by Reviewer 1 (recreated below), we also explain why we believe experiments on humans to establish this fact is not yet reasonable.

First, we discuss the mock finger. The PDMS finger is treated to have comparable surface and bulk properties to a human finger. We have approximated the softness of the finger with 100 kPa crosslinked PDMS, which is close to what has been reported for the bulk of a human fingertip.[9,10] However, as mentioned in the Materials and Methods, there are two additional features of the mock finger that impart greater strength. The PDMS surrounds a rigid, acrylic bone comparable to the distal phalanx, which provides an additional layer of higher modulus.[8] Additionally, the 8-hour UV-Ozone treatment decreases the viscoelastic tack of the pristine PDMS by glassifying, or further crosslinking the surface of the finger,[12] therefore imparting greater stiffness at the surface similar to the contributions of the stratum corneum, along with a similar surface energy.[13] Additionally, the finger is used at least a day after UV-Ozone treatment is completed in order for the surface to return to moderate hydrophilicity, similar to the outermost layer of human skin.[17] We also discuss the shape of the contact formed. To ensure that there is minimal contribution from the slanted position of the finger, an initial contact area of 1×1 cm is established before sliding and recording friction measurements. As the PDMS finger is a soft object, the portion in contact with a surface flattens and the contact area remains largely unchanged during sliding. Any additional stick-slip after this alignment step is caused by contact aging at the interface, but the first trace we collect is always discarded to only consider stick-slip events caused by surface chemistry. We recognize that it is difficult to completely control the pressure distribution due to the planar interface, but this is also expected when humans freely explore a surface. Finally, we consider flat vs. fingerprinted fingers. Our previous work on the role of fingerprints on friction experienced by a PDMS mock finger showed enhanced signals with the incorporation of ridges on the finger and used a rate-and-state model of a heterogenous, elastic body to find corresponding trends.[11] The key conclusion was that a flat finger still preserved key dynamic features, and the presence of stronger or more vibrations could result in more similar forces for different surfaces depending on the sliding conditions. We note that we have subsequently used this flat mock finger in correlations with human psychophysics in previous work, where findings from our mechanical experiments were predictive of human performance.[4–7] We have added these details to the manuscript.

With this adequately similar mock finger, we collected friction traces at controlled conditions of normal force and velocity in order to extract the signals unique to each material that are not caused by the influence of human variability. For example, we observe the smallest regions of steady sliding on our phase maps (Fig. 1C) for short-chain alkylsilanes C4 and C5, while the increased intermolecular forces of other silanes increase the incidence of steady sliding. We have also previously shown that comparisons of similarly collected mechanical data is predictive of human performance, using the crosscorrelations between signals of two different materials.[4–7] While different participants produce different raw signals, we see that broad categories of stick-slip, i.e. instabilities, can be extracted (Fig. 3B-C) and used as a cue in a tactile discrimination task. As mentioned above, we have provided an additional section about the usefulness of our mock finger, as well as its structure, in the main manuscript.

Second, we lay out the challenges and barriers to demonstrating this in humans in the manner requested by the reviewer, and how the scope of this paper will lead us in that direction. We also clarify that our goals are to provide a method to characterize samples to better design tactile interfaces in haptics or in psychophysical experiments and raise awareness that the common methods of sample characterization in touch by an average friction coefficient or roughness is fundamentally unsound.

As discussed in the summary, and with additional detail here, to further support our findings through observation on humans would require answering:

(1) Which one, or combination of, of the multiple swipes that people make responsible for a tactile decision?

(2) Establish what is, or may be, tactile evidence.

(3) Establish tactile decision-making models are similar or different than existing decision-making models.

(4) Test the hypothesis, in these models, that friction instabilities are evidence, and not some other unknown metric.

(5) Design a task that does not require the use of subjective tactile descriptors, like “which one feels rougher”, which we see cause confusion in participants, which will likely require accounting for memory effects.

We elaborate these points below:

To successfully perform this experiment, we note that freely exploring humans make multiple strokes on a surface. Therefore, we would need to construct a decision-making model. It has not yet been demonstrated whether tactile decision making follows visual decision making, but perhaps to start, we can assume it does. Then, in the design of our decision-making paradigm, we immediately run into the problem: What is tactile evidence?

From Fig. 3C, we already can see that identifying evidence is challenging. Prior to this manuscript, people may have chosen the average force, or the highest force. Or we may choose the average friction force. Then, after deciding on the evidence, we need to find a method to manipulate the evidence, i.e., create samples or a machine that causes high friction, etc. We show that during the course of human touch, due to the dynamic nature of friction, the average can change a large amount and sample design becomes a central barrier to experiments. Others may suggest immobilizing the finger and applying a known force, but given how much friction changes with human exploration, there is no known method to make a machine recreate temporally and spatially varying friction forces during sliding onto a stationary finger. Finally, perhaps most importantly, in addition to mechanical challenges, a study by Liu, Colgate, et al. showed that even if they recorded the friction (2D) of a finger exploring a surface and then replicated the same friction forces onto a finger, the participant could not determine which surface the replayed friction force was supposed to represent.[1] This supports that the efference copy is important, that the forces in response to expected motion are important to determine friction. Finally, there is no known method to design instabilities *a priori*. They must be found through experiments, especially since if we were to introduce, say a bump or a trough, then we bring in confounding variables to how participants tell surfaces apart.

Furthermore, even if we had some consistent method to create tactile “evidence”, the paradigm also deserves some consideration. In our experience, the 3-AFC task we perform is important because the vocabulary for touch has not been established. That is, in 3-AFC, by asking to determine which one sample is unlike the others, we do not have to ask the participant questions like “which one is rougher” or “which one has less friction”. In contrast, 2-AFC, which is better for decision-making models because it does not include memory, requires the asking of a perceptual question like: “which one is rougher?”. In our ongoing work, taking two silane coatings, we found that participants could easily identify which surface is unlike the others above chance in a 3-AFC, but participants, even within their own trials, could not consistently identify one silane as perceptually “rougher” by 2-AFC. To us, this calls into question the validity of tactile descriptors, but is beyond the scope of the current manuscript.

This is not our only goal, but in the context of human exploration, in this manuscript here, we believed it was important to identify a mechanical parameter that was consistent with how humans explore surfaces, but was also a parameter that could characterize to some consistent property of a surface – irrespective of whether a human was touching it. We thought that designing human decision-making models and paradigms around the friction coefficient would not be successful.

Given the scope of these challenges, we do not think it would be possible to establish this conceptual sequence in a single manuscript.

See Reviewer 1, comment 3part 3 for changes to the manuscript

Comment 2I believe that the authors collected the contact forces during the psychophysics experiments, so this shortcoming could be solved if the authors use the actual data, and show that the participant responses can be better predicted by the occurrence of frictional instabilities than by the usual metrics on a trial by trial basis, or at least on a subject by subject basis. I.e. Poor performers should show fewer signs of differences in the sliding behaviors than good performers.

To fully implement this, a decision-making model is necessary because, as a counter example, a participant could have generated 10 swipes of SFW and 1 swipe of a Sp, but the Sp may have been the most important event for making a tactile decision. This type of scenario is not compatible with the analysis suggested — and similar counterpoints can be made for other types of seemingly straightforward analysis.

While we are interested and actively working on this, the study here is critical to establish types of evidence for a future decision-making model. We know humans change their friction constantly during real exploration, so it is unclear which of these constantly changing values we should input into the decision making model, and the future challenges we anticipate are explained in Weaknesses, Comment 1.

Comment 3The sample size (10) is very small.

We recognize that, with all factors being equal, this sample size is on the smaller end. However, we emphasize the degree of control of samples is far above typical, with minimal variations in sample properties such as surface roughness, and every sample for every trial was pristine. Furthermore, the sample preparation (> 300 individual wafers were used) became a factor. Although not typically appropriate, and thus not included in the manuscript, a post-hoc power analysis for our 100 trials of our pair that was closest to chance, *P4*, (53%, closest to chance at 33%) showed a power of 98.2%, suggesting that the study was appropriately powered.

**Reviewer 2 (Recommendations for the authors):**
Comment 1Differences in SS and Sp (Table 2) are NOT physical or mechanical differences but are obtained by counting differences in the number of occurrences of each sliding behavior. It is rather a weird choice.

We disagree that differences in SS and Sp are not physical or mechanical, as these are well-established phenomena in the soft matter and tribology literature.[20–22] These are known as “mechanical instabilities” and generated due to the effects of two physical phenomena: the elasticity of the finger (which is constant in our mechanical testing) and the friction forces present (which change per sample type). The motivation behind using these different shapes is that the instabilities, in some conditions, can be invariant to external factors like velocity. This would be quite advantageous for human exploration because, unlike friction coefficient, which changes with nearly any factor, including velocity and mass, the instabilities being invariant to velocity would mean that we are accurately characterizing a unique identifier of the surface even though velocity may be variable.

This “weird choice” is the central innovation of this paper. This choice was necessary because we demonstrated that the common usage of friction coefficient is fundamentally flawed: we see that friction coefficient suggests that surface which are more different would feel more similar – indeed the most distinctive surfaces would be two surfaces that are identical, which is clearly spurious. Furthermore, Table 1 now includes the range of friction generated on a surface, the range of friction coefficients of a single surface is large – of order the differences in friction between two surfaces. This is expected in soft sliding systems and emphasizes our issue with the use of average friction coefficient in psychophysical design. One potential explanation for why we were able to see this is effect is because our surfaces have similar (< 0.6 nm variability) roughness, removing potential confounding factors from large scale roughness, and this type of low roughness control has not been widely used in tactile studies to the best of our knowledge.

Comment 2Figures 2B-C: why are the x-data different than Table 2?

The x-data in Fig. 2B-C are the absolute differences in the number of occurrences measured for a given instability type or material property out of 144 pulls. Modeling the human participant results in our GLMMs required the independent variables to be in this form rather than percentages. We initially chose to list percent differences in Table 2 to highlight the ranges of differences instead of an absolute value, but have added both for clarity.

Our changes to the manuscript

Page 7

“To determine if humans can detect these three different instabilities, we selected six pairs of surfaces to create a broad range of potential instabilities present across all three types. These are summarized in Table 2, where the first column for each instability is the difference in occurrence of that instability formed between each pair, and the second is the percent difference.”

“Thus, when comparing C4 versus C4-APTMS, they have a difference in steady sliding of 20 out of a maximum 144 pulls, for a |ΔSS| of 13.9%. The absolute value is taken to compare total differences present, as the psychophysical task does not distinguish between sample order.”

Comment 3We constructed a set of coated surfaces with physical differences which were imperceptible by touch but created different types of instabilities based on how quickly a finger is slid and how hard a human finger is pressed during sliding." Yet, in your experiment, participants could discriminate them, so this is incoherent.

To clarify the point, macroscopic objects can differ in physical shape and in chemical composition. What we meant was that the physical differences, i.e., roughness, were below a limit (Skedung et al.) that participants, without a coating, would not be able to tell these apart.[23] Therefore, the reason people could tell our surfaces apart was due to the chemical composition of the surface, and not any differences in roughness or physical effects like film stiffness (due to the molecular-scale thinness of the surface coatings, they are mechanically negligible). However, we concede that at the molecular scale, the traditional macroscopic distinction between physical and chemical is blurred.

We have made minor revisions to the wording in the abstract. We clarify that the surface coatings had physical differences in roughness that were smaller than 0.6 nm, which based purely on roughness, would not be expected to be distinguishable to participants. Therefore, the reason participants can tell these surfaces apart is due to differences in friction generated by chemical composition, and we were able to minimize contributions from physical differences in the sample our study.

Our changes to the manuscript

Page 1, Abstract

“Here, we constructed a set of coated surfaces with minimal physical differences that by themselves, are not perceptible to people, but instead, due to modification in surface chemistry, the surfaces created different types of instabilities based on how quickly a finger is slid and how hard a human finger is pressed during sliding.”

“In one experiment, we used a mechanical mock finger to quantify and classify differences in instability formation from different coated surfaces. In a second experiment, participants perform a discrimination task using the same coated surfaces. Using the data from these two experiments, we found that human discrimination response times were faster with surfaces where the mock finger produced more stiction spikes and discrimination accuracy was higher where the mock finger produced more steady sliding. Conversely, traditional metrics like surface roughness or average friction coefficient did not relate to tactile discriminability. In fact, the typical method of averaging friction coefficients led to a spurious correlation which erroneously suggests that distinct objects should feel identical and identical objects should feel distinct—similar to findings by others. Friction instabilities may offer a more predictive and tractable framework of fine touch perception than friction coefficients, which would accelerate the design of tactile interfaces.”

**Reviewer 3 (Public review):**
StrengthsThe paper describes a new perspective on friction perception, with the hypothesis that humans are sensitive to the instabilities of the surface rather than the coefficient of friction. The paper is very well written and with a comprehensive literature survey.One of the central tools used by the author to characterize the frictional behavior is the frictional instabilities maps. With these maps, it becomes clear that two different surfaces can have both similar and different behavior depending on the normal force and the speed of exploration. It puts forward that friction is a complicated phenomenon, especially for soft materials.The psychophysics study is centered around an odd-one-out protocol, which has the advantage of avoiding any external reference to what would mean friction or texture for example. The comparisons are made only based on the texture being similar or not.The results show a significant relationship between the distance between frictional maps and the success rate in discriminating two kinds of surface.

We thank Reviewer 3 for their notes and interesting discussion points on our manuscript. Below, we address the reviewer’s feedback and comments on related works.

Weaknesses:Comment 1The main weakness of the paper comes from the fact that the frictional maps and the extensive psychophysics study are not made at the same time, nor with the same finger. The frictional maps are produced with an artificial finger made out of PDMS which is a poor substitute for the complex tribological properties of skin.

A similar comment was made by Reviewers 1 and 2. We agree in part and have made changes throughout that our study is correlative, but presents an important step forward to these biomechanical measurements and corresponding decision making models.

We are not claiming that our PDMS fingers are superior to real fingers, but rather, we cannot establish standards in the field by using real human fingers that vary between subjects and researchers. We believe the mock finger we designed is a reasonable mimic of the human finger by matching surface energy, heterogeneous mechanical structure, and the ability to test multiple physiologically relevant pressures and sliding velocities.

We achieve a heterogeneous mechanical structure with the 3 primary components of stiffness of a human finger. The effective modulus of ~100 kPa, from soft tissue,[9,10] is obtained with a 30:1 ratio of PDMS to crosslinker. The PDMS also surrounds a rigid, acrylic bone comparable to the distal phalanx, which provides an additional layer of higher modulus.[8] Additionally, the 8-hour UV-Ozone treatment decreases the viscoelastic tack of the pristine PDMS by glassifying, or further crosslinking the surface of the finger,[12] therefore imparting greater stiffness at the surface similar to the contributions of the stratum corneum, along with a similar surface energy.[13] The finger is used at least a day after UV-Ozone treatment is completed in order for the surface to return to moderate hydrophilicity, similar to the outermost layer of human skin.[17] We also discuss the shape of the contact formed. To ensure that there is minimal contribution from the slanted position of the finger, an initial contact area of 1×1 cm is established before sliding and recording friction measurements. As the PDMS finger is a soft object, the portion in contact with a surface flattens and the contact area remains largely unchanged during sliding. We recognize that it is difficult to completely control the pressure distribution due to the planar interface, but this variation is also expected when humans freely explore a surface. Finally, we consider flat vs. fingerprinted fingers. Our previous work on the role of fingerprints on friction experienced by a PDMS mock finger showed enhanced signals with the incorporation of ridges on the finger and used a rate-andstate model of a heterogenous, elastic body to find corresponding trends.[11] The key conclusion was that a flat finger still preserved key dynamic features, and the presence of stronger or more vibrations could result in more similar forces for different surfaces depending on the sliding conditions. We note that we have subsequently used the controlled mechanical data collected with this flat mock finger in correlations with human psychophysics in previous work, where findings from our mechanical experiments were predictive of human performance.[4–7] Ultimately, we see from our prior work and here that, despite the drawbacks of our mock finger, it outperforms other standard characterization technique in providing information about the mesoscale that correlates to tactile perception. We have added these details to the manuscript.

We also note that an intermediate option, replicating real fingers, even in a mold, may also inadvertently limit trends from characterization to a specific finger. One of the main – and severe – limitations of using a human finger is that all fingers are different, meaning any study focusing on a particular user may not apply to others or be recreated easily by other researchers. We cannot set a standard for replication around a real human finger as that participant may no longer be available, or willing to travel the world as a “standard”. Furthermore, the method in which a single person changes their pressures and velocities as they touch a surface is highly variable. We also note that in the Summary Response, we noted that a study by Colgate et al. (IEEE ToH 2024) demonstrated that efference copies may be important, and thus constraining a human finger and replaying the forces recorded during free exploration will not lead to the participant identifying a surface with any consistency. Thus, it is important to allow humans to freely explore surfaces, but creates nearly limitless variability in friction forces.

This is also against the backdrop that we are seeking to provide a method to characterize surfaces. Indeed, the more features we replicate in the mock finger to a human finger, the more likely it is that the mechanical data will correlate to human performance. However, we have used this technique several times to achieve stronger correlations to human data than other available techniques. We believe the metric of success should be in comparison to the available characterization technique, rather than a 1:1 reconstruction of forces of an arbitrary human finger. Indeed, a 1:1 reconstruction of forces of an arbitrary human finger would be limited to the finger of a single individual, perhaps even to that individual on a given day.

See Reviewer1 weaknesses, comment 2 part 2 for changes to the manuscript

Comment 2The evidence would have been much stronger if the measurement of the interaction was done during the psychophysical experiment. In addition, because of the protocol, the correlation is based on aggregates rather than on individual interactions.

We agree that this would have helped further establish our argument, but in the overall statement and in other reviewer responses, we describe the significant challenges to establishing this.

To fully implement this, a decision-making model is necessary because, as a counter example, a participant could have generated 10 swipes of SFW and 1 swipe of a Sp, but the Sp may have been the most important event for making a tactile decision. We also clarify that our goals are to provide a method to characterize samples to better design tactile interfaces in haptics or in psychophysical experiments.

As discussed in the summary, and expanded on here, in our view, to develop a decision-making model, the challenges are as follows:

(1) Which one, or combination of, of the multiple swipes that people make responsible for a tactile decision?

(2) Establish what is, or may be, tactile evidence.

(3) Establish tactile decision-making models are similar or different than existing decision-making models.

(4) Test the hypothesis, in these models, that friction instabilities are evidence, and not some other unknown metric.

(5) Design a task that does not require the use of subjective tactile descriptors, like “which one feels rougher”, which we see cause confusion in participants, which will likely require accounting for memory effects.

(6) Design samples that vary in the amount of evidence generated, but this evidence cannot be controlled directly. Rather, the samples indirectly vary evidence by how likely it is for a human to generate different types of friction instabilities during standard exploration.

We elaborate these points below:

To successfully perform this experiment, we note that freely exploring humans make multiple strokes on a surface. Therefore, we would need to construct a decision-making model. It has not yet been demonstrated whether tactile decision making follows visual decision making, but perhaps to start, we can assume it does. Then, in the design of our decision-making paradigm, we immediately run into the problem: What is tactile evidence?

From Fig. 3C, we already can see that identifying evidence is challenging. Prior to this manuscript, people may have chosen the average force, or the highest force. Or we may choose the average friction force. Then, after deciding on the evidence, we need to find a method to manipulate the evidence, i.e., create samples or a machine that causes high friction, etc. We show that during the course of human touch, due to the dynamic nature of friction, the average can change a large amount and sample design becomes a central barrier to experiments. Others may suggest to immobilize the finger and applying a known force, but given how much friction changes with human exploration, there is no known method to make a machine recreate temporally and spatially varying friction forces during sliding onto a stationary finger. Finally, perhaps most importantly, in addition to mechanical challenges, a study by Liu, Colgate et al. showed that even if they recorded the friction (2D) of a finger exploring a surface and then replicated the same friction forces onto a finger, the participant could not determine which surface the replayed friction force was supposed to represent.[1] This supports that the efference copy is important, that the forces in response to expected motion are important to determine friction. Finally, there is no known method to design instabilities *a priori*. They must be found through experiments, especially since if we were to introduce, say a bump or a trough, then we bring in confounding variables to how participants tell surfaces apart.

Furthermore, even if we had some consistent method to create tactile “evidence”, the paradigm also deserves some consideration. In our experience, the 3-AFC task we perform is important because the vocabulary for touch has not been established. That is, in 3-AFC, by asking to determine which one sample is unlike the others, we do not have to ask the participant questions like “which one is rougher” or “which one has less friction”. In contrast, 2-AFC, which is better for decision-making models because it does not include memory, requires the asking of a perceptual question like: “which one is rougher?”. In our ongoing work, taking two silane coatings, we found that participants could easily identify which surface is unlike the others above chance in a 3-AFC, but participants, even within their own trials, could not consistently identify one silane as perceptually “rougher” by 2-AFC. To us, this calls into question the validity of tactile descriptors, but is beyond the scope of the current manuscript.

This is not our only goal, but in the context of human exploration, in this manuscript here, we believed it was important to identify a mechanical parameter that was consistent with how humans explore surfaces, but was also a parameter that could characterize to some consistent property of a surface – irrespective of whether a human was touching it. We thought that designing human decision-making models and paradigms around the friction coefficient would not be successful.

Given the scope of these challenges, we do not think it would be possible to establish this conceptual sequence in a single manuscript.

Comment 3The authors compensate with a third experiment where they used a 2AFC protocol and an online force measurement. But the results of this third study, fail to convince the relation.

With this experiment, our central goal was to demonstrate that the instabilities we have identified with the PDMS finger also occur with a human finger. Several instances of SS, Sp, and SFW were recorded with this setup as a participant touched surfaces in real time.

Comment 4No map of the real finger interaction is shown, bringing doubt to the validity of the frictional map for something as variable as human fingers.

Real fingers change constantly during exploration, and friction is state-dependent, meaning that the friction will depend on how the person was moving the moment prior. Therefore, a map is only valid for a single human movement – even if participants all were instructed to take a single swipe and start from zero motion, humans are unable to maintain constant velocities and pressures. Clearly, this is not sustainable for any analysis, and these drawbacks apply to any measured parameter, whether instabilities suggested here, or friction coefficients used throughout. We believe the difficulty of this approach emphasizes why a standard map of characterization of a surface by a mock finger, even with its drawbacks, is a viable path forward.

**Reviewer 3 (Recommendations for the authors):**
Comment 1It would be interesting to comment on a potential connection between the frictional instability maps and Schalamack waves.

Schallamach waves are a subset of slow frictional waves (SFW). Schallamach waves are very specifically defined in the field. They occur when pockets of air that form between a soft sliding object and rigid surface which then propagate rear-to-front (retrograde waves) relative to motion of the sliding motion and form buckles due to adhesive pinning. Wrinkles then form at the detached portion of the soft material, until the interface reattaches and the process repeats.[24] There is typically a high burden of proof to establish a Schallamach wave over a more general slow frictional wave. We note that it would be exceedingly difficult to design samples that can reliably create subsets of SFW, but we are aware that this may be an interesting question at a future point in our work.

Comment 2The force sensors look very compliant, and given the dynamic nature of the signal, it is important to characterize the frequency response of the system to make sure that the fluctuations are not amplified.

Thank you for noticing. We mistyped the sensor spring constant as 13.9 N m^-1^ instead of kN m^-1^. However, below we show how the instabilities are derived from the mechanics at the interface due to the compliance of the finger. The “springs” of the force sensor and PDMS finger are connected in parallel. Since *ksensor* = 13.9 kN m^-1^, the spring constant of the system overall reflects the compliance of the finger, and highlights the oscillations arising solely from stick-slip. A sample calculation is shown below.

Author response image 1.

Fitting a line to the initial slope of the force trace for C6 gives the equation y = 25.679x – 0.2149. The slope here represents force data over time data, and is divided by the velocity (25 mm/s) to determine the spring constant of the system *ktotal* = \begin{document}$\frac{F}{x}$\end{document} = 1027.16 N/m. This value is lower than *ksensor* = 13.9 kN/m, indicating that the “springs” representing the force sensor and PDMS finger are connected in parallel:

\begin{document}$\frac{1}{k_{\text {total }}}=\frac{1}{k_{\text {sensor }}}+\frac{1}{k_{\text {finger }}}$\end{document}. The finger is the compliant component of the system, with *kfinger* = 1.11 kN/m, and of course, real human fingers are also compliant so this matches our goals with the design of the mock finger.\begin{document}$$\displaystyle 25 \mathrm{~mm} / \mathrm{s}, 25 \mathrm{~g}$$\end{document}

Our changes to the manuscript

(Page 4) (*k* = 13.9 kN m^1^)

Comment 3The authors should discuss about the stochastic nature of friction: - Wiertlewski, Hudin, Hayward, IEEE WHC 2011 Greenspon, McLellan, Lieber, Bensmaia, JRSI 2020.

We believe that, given the references, this comment on “stochastic” refers to the macroscopically-observable fluctuations (i.e., the mechanical “noise” which is not due to instrument noise) in friction arising from the discordant network of stick-slip phenomena occurring throughout the contact zone, and not the stochastic nature of nanoscale friction that occurs thermal fluctuations nor due to statistical distributions in bond breaking associated with soft contact.

We first note that our small-scale fluctuations do not arise from a periodic surface texture that dominates in the frequency regime. However, even on our comparatively smooth surfaces, we do expect fluctuations due to nanoscale variation in contact, generation of stick-slip across at microscale length scales that occur either concurrently or discordantly across the contact zone, and the nonlinear dependence of friction to nearly any variation in state and composition.[11]

Perhaps the most relevant to the manuscript is that a major advantage of analysis by friction is that it sidesteps these ever-present microscale fluctuations, leading to more clearly defined classifiers or categories during analysis. Wiertlewski et. al. showed repeated measurements in their systems ultimately gave rise to consistent frequencies[25] (we think their system was in a steady sliding regime and the patterning gave rise to underlying macroscopic waves). These consistent frequencies, at least in soft systems and absent obvious macroscopic patterned features, would be expected to arise from the instability categories and we see them throughout.

Comment 4It is stated that "we observed a spurious, negative correlation between friction coefficient and accuracy".What makes you qualify that correlation as spurious?

We mean this as in the statistical definition of “spurious”.

This correlation would indicate that by the metric of friction coefficient, more different surfaces are perceived more similarly. Thus, two very different surfaces, like Teflon and sandpaper, by friction coefficient would be expected to feel very similar. Two nearly identical surfaces would be expected to feel very different – but of course, humans cannot consistently distinguish two identical surfaces. This finding is counterintuitive and refutes that friction coefficient is a reliable classifier of surfaces by touch. We do not think it is productive to determine a mechanism for a spurious correlation, but perhaps one reason we were able to observe this is because our study, to the best of our knowledge, is unique for having samples that are controlled in their physical differences in roughness and surface features.

See response to Reviewer 1 weaknesses, comment 1 for changes to the manuscript

Comment 5The authors should comment on the influence of friction on perceptual invariance. Despite inducing radially different frictional behavior for various conditions, these surfaces are stably perceived. Maybe this is a sign that humans extract a different metric?

We agree – we are excited that frictional instabilities may offer a more stable perceptual cue because they are not prone to fluctuations (as discussed in Comment 3) and instability formation, in many conditions, is invariant to applied pressures and velocities – thus forming large zones where a human may reasonable encounter a given instability.

Raw friction is highly prone to variation during human exploration (in alignment with Recommendations for the authors, Comment 3), but ongoing work seeks to explain tactile constancy, or the ability to identify objects despite these large changes in force. Very recently published work by Fehlberg et. al. identified the role of modulating finger speed and normal force in amplifying the differences in friction coefficient between materials in order to identify them,[26] and we postulate that their work may be streamlined and consistent with the idea of friction instabilities, though we have not had a chance to discuss this in-depth with the authors yet.

We think that the instability maps show a viable path forward to how surfaces are stably perceived, and instabilities themselves show a potential mechanism: mathematically, instabilities for given conditions can be invariant to velocity or mass, creating zones where a certain instability is encountered. This reduces the immense variability of friction to a smaller, more stable classification of surfaces (e.g., a 30% SS surface or a 60% SS surface). A given surface will typically produce the same instability at a specific condition (we found some boundaries of experimental parameters are very condition sensitive, but many conditions are not), whereas a single friction trace which is highly prone to variation is not a stable metric.

Added Reference

(53) M. Fehlberg, E. Monfort, S. Saikumar, K. Drewing and R. Bennewitz, *IEEE Trans. Haptics*, 2024, 17, 957–963.

References

(1) Liu, Z., Kim, J.-T., Rogers, J. A., Klatzky, R. L. & Colgate, J. E. Realism of Tactile Texture Playback: A Combination of Stretch and Vibration. IEEE Trans. Haptics 17, 441–450 (2024).

(2) Waters, I., Alazmani, A. & Culmer, P. Engineering Incipient Slip Into Surgical Graspers to Enhance Grasp Performance. IEEE Transactions on Medical Robotics and Bionics 2, 541–544 (2020).

(3) Gueorguiev, D., Bochereau, S., Mouraux, A., Hayward, V. & Thonnard, J.-L. Touch uses frictional cues to discriminate flat materials. Sci Rep 6, 25553 (2016).

(4) Carpenter, C. W. et al. Human ability to discriminate surface chemistry by touch. Mater. Horiz. 5, 70– 77 (2018).

(5) Nolin, A. et al. Predicting human touch sensitivity to single atom substitutions in surface monolayers for molecular control in tactile interfaces. Soft Matter 17, 5050–5060 (2021).

(6) Nolin, A. et al. Controlling fine touch sensations with polymer tacticity and crystallinity. Soft Matter 18, 3928–3940 (2022).

(7) Swain, Z. et al. Self-Assembled Thin Films as Alternative Surface Textures in Assistive Aids with Users Who are Blind. J. Mater. Chem. B (2024) doi:10.1039/D4TB01646G.

(8) Qian, K. et al. Mechanical properties vary for different regions of the finger extensor apparatus. J Biomech 47, 3094–3099 (2014).

(9) Abdouni, A. et al. Biophysical properties of the human finger for touch comprehension: influences of ageing and gender. Royal Society Open Science (2017) doi:10.1098/rsos.170321.

(10) Cornuault, P.-H., Carpentier, L., Bueno, M.-A., Cote, J.-M. & Monteil, G. Influence of physicochemical, mechanical and morphological fingerpad properties on the frictional distinction of sticky/slippery surfaces. Journal of The Royal Society Interface (2015) doi:10.1098/rsif.2015.0495.

(11) Dhong, C. et al. Role of fingerprint-inspired relief structures in elastomeric slabs for detecting frictional differences arising from surface monolayers. Soft Matter 14, 7483–7491 (2018).

(12) Fu, Y.-J. et al. Effect of UV-Ozone Treatment on Poly(dimethylsiloxane) Membranes: Surface Characterization and Gas Separation Performance. Langmuir 26, 4392–4399 (2010).

(13) Yuan, Y. & Verma, R. Measuring microelastic properties of stratum corneum. Colloids Surf B Biointerfaces 48, 6–12 (2006).

(14) Yu, G. et al. A wearable pressure sensor based on ultra-violet/ozone microstructured carbon nanotube/polydimethylsiloxane arrays for electronic skins. Nanotechnology 29, 115502 (2018).

(15) Zheng, L. et al. Dual-Stimulus Smart Actuator and Robot Hand Based on a Vapor-Responsive PDMS Film and Triboelectric Nanogenerator. ACS Appl. Mater. Interfaces 11, 42504–42511 (2019).

(16) Ma, K., Rivera, J., Hirasaki, G. J. & Biswal, S. L. Wettability control and patterning of PDMS using UV–ozone and water immersion. Journal of Colloid and Interface Science 363, 371–378 (2011).

(17) Mavon, A. et al. Sebum and stratum corneum lipids increase human skin surface free energy as determined from contact angle measurements: A study on two anatomical sites. Colloids and Surfaces B: Biointerfaces 8, 147–155 (1997).

(18) AliAbbasi, E. et al. Effect of Finger Moisture on Tactile Perception of Electroadhesion. IEEE Trans. Haptics 17, 841–849 (2024).

(19) Corniani, G. et al. Sub-surface deformation of individual fingerprint ridges during tactile interactions.

eLife 13, (2024).

(20) Israelachvili, J. N. Intermolecular and Surface Forces. (Academic Press, 2011).

(21) Das, S. et al. Stick–slip friction of gecko-mimetic flaps on smooth and rough surfaces. J R Soc Interface 12, 20141346 (2015).

(22) Persson, B. N. J., Albohr, O., Creton, C. & Peveri, V. Contact area between a viscoelastic solid and a hard, randomly rough, substrate. The Journal of Chemical Physics 120, 8779–8793 (2004).

(23) Skedung, L. et al. Feeling Small: Exploring the Tactile Perception Limits. Sci Rep 3, 2617 (2013).

(24) Viswanathan, K., Sundaram, N. K. & Chandrasekar, S. Stick-slip at soft adhesive interfaces mediated by slow frictional waves. Soft Matter 12, 5265–5275 (2016).

(25) Wiertlewski, M., Hudin, C. & Hayward, V. On the 1/f noise and non-integer harmonic decay of the interaction of a finger sliding on flat and sinusoidal surfaces. in 2011 IEEE World Haptics Conference 25–30 (2011). doi:10.1109/WHC.2011.5945456.

(26) Fehlberg, M., Monfort, E., Saikumar, S., Drewing, K. & Bennewitz, R. Perceptual Constancy in the Speed Dependence of Friction During Active Tactile Exploration. IEEE Transactions on Haptics 17, 957–963 (2024).